



# The European 2015 drought from a groundwater perspective: estimation in absence of observed groundwater data

Anne F. Van Loon[1,*], Rohini Kumar[2,*], and Vimal Mishra[3]

[*]These authors contributed equally to this work
[1]School of Geography, Earth and Environmental Sciences, University of Birmingham, UK
[2]UFZ-Helmholtz Centre for Environmental Research, Leipzig, Germany
[3]Civil Engineering, Indian Institute of Technology, Gandhinagar, Gujarat, India

*Correspondence to:* Anne Van Loon (a.f.vanloon@bham.ac.uk)

**Abstract.** In 2015, central and eastern Europe were affected by a severe drought. This event has recently been studied from meteorological and streamflow perspective, but no analysis of the groundwater situation has been performed. One of the reasons is that real-time groundwater level observations often are not available. In this study, we use spatially-explicit relationships between meteorological conditions and historic groundwater level observations to quantify the 2015 groundwater drought over two regions in southern Germany and eastern Netherlands. We use the monthly groundwater observations from 2040 wells to establish the spatially-varying optimal accumulation period between the Standardised Groundwater Index (SGI) and the Standardised Precipitation Evapotranspiration Index (SPEI) at a 0.25 gridded scale. The resulting optimal accumulation periods range between 1 and more than 24 months, indicating strong spatial differences in groundwater response time to meteorological input over the region. The Standardised Precipitation Index (SPI) showed similar results, which point to a limited influence of potential evaporation in determining the period of influence of meteorological conditions on groundwater levels in our study regions. Based on the estimated optimal accumulation periods and available meteorological time series, we reconstructed the groundwater anomalies up to 2015 and found that in Germany a uniform severe groundwater drought persisted for several months in this year, whereas the Netherlands appeared to have relatively high groundwater levels. The differences between this event and the 2003 European benchmark drought are striking. The 2003 groundwater drought was less uniformly pronounced, both in the Netherlands and Germany. This is because slowly responding wells (the ones with optimal accumulation periods of more than 12 months) still were above average from the wet year of 2002, which experienced severe flooding in central Europe. We also tested the applicability of the Gravity Recovery Climate Experiment (GRACE) Terrestrial Water Storage (TWS) and GRACE-based groundwater anomalies to capture the spatial variability of the 2003 and 2015 drought events. GRACE-TWS does show that both 2003 and 2015 were relatively dry, but the difference between Germany and the Netherlands in 2015 and the spatially-variable groundwater drought pattern in 2003 were not captured. This could be associated to the coarse spatial scale of GRACE. The simulated groundwater anomalies based on GRACE-TWS deviate considerably from the GRACE-TWS signal and from observed groundwater anomalies. These are therefore not suitable for use in real-time groundwater drought monitoring in our case study regions. Our novel approach based on the spatially-variable relationship between meteorological conditions and groundwater levels allow to quantify groundwater drought in near real-time. We found that the 2015 groundwater drought in southern Germany was more severe than the 2003 drought, because of preconditions in





slowly responding aquifers. Compared to the meteorological drought and streamflow drought (described in previous studies), the groundwater drought of 2015 had a more pronounced spatial variability in its response to meteorological conditions, with some areas primarily influenced by short-term meteorological deficits and others influenced by meteorological deficits accumulated over the preceding two years or more. In drought management, this information is very useful and our approach 5 to quantify groundwater drought can be used until real-time groundwater observations become readily available.

# 1 Introduction

In the summer of 2015 large parts of Europe were affected by a severe drought (Van Lanen et al., 2016; Orth et al., 2016). This drought event has been analysed from climatological (Ionita et al., 2016; Orth et al., 2016) and hydrological (Laaha et al., 2016) perspective, which gives a useful overview of the causes, development, extent and severity of the drought event. 10 Ionita et al. (2016), for example, found that the below-normal precipitation amounts in the summer of 2015 were related to positive geopotential height anomalies over Europe, large negative SST anomalies in the North-Atlantic Ocean and positive SST anomalies in the Mediterranean Sea. The effects of the below-normal precipitation and high temperatures during summer 2015 resulted in streamflow droughts with return periods of 100 years and more in central Europe (Laaha et al., 2016). Impacts of the drought were felt across many sectors, including agriculture, drinking water supply, electricity production, navigation, 15 fisheries, and recreation (Van Lanen et al., 2016).

Van Lanen et al. (2016) and Laaha et al. (2016) note that hydrological information is key to understanding and managing the impacts of drought events and that hydrological drought needs to be monitored. Hydrological drought encompasses both below normal river flow and below normal groundwater levels (Van Loon, 2015). For the 2015 event, however, Laaha et al. (2016) could only analyse streamflow drought because of the absence of near real-time groundwater level data for Europe. Monitoring 20 groundwater drought, however, is highly relevant for some drought-impacted sectors such as water supply and irrigation. Additionally, groundwater is linked to other hydrological variables such as soil moisture and streamflow; and monitoring groundwater during drought gives important information about drought persistence in these related variables.

Timely groundwater drought analysis is often hampered by a lack of real-time data. Groundwater drought studies are therefore often done on historic data sets (e.g., Chang and Teoh, 1995; Peters et al., 2006; Tallaksen et al., 2009; Shahid and Hazarika, 25 2010; Bloomfield and Marchant, 2013; Kumar et al., 2016). For adequate groundwater drought monitoring and early-warning, however more actual information is essential (Bachmair et al., 2016). Up to date, different approaches have been explored that aim to quantify groundwater drought in the absence of near real-time groundwater level data, using either long-term precipitation, streamflow, land surface models or GRACE satellites data (e.g., Thomas et al., 2015; Laaha et al., 2016; Li and Rodell, 2015; Thomas et al., 2014).

30 An approach that is often used is accumulating precipitation anomalies over a long period that is assumed to be representative for groundwater fluctuation (Zeng et al., 2008; Raziei et al., 2008; Thomas et al., 2015). The assumption is that a precipitation-based index (like SPI) with a long accumulation period can adequately represent anomalies in groundwater level. Good correlations are found when groundwater time series of a high number of wells are averaged over a large region or





when only annual averages are considered (Joetzjer et al., 2013). Although groundwater on average responds to precipitation on long timescales, several studies have found that groundwater drought often has a heterogeneous pattern and can hardly be represented by a spatially uniform accumulation period (e.g., 6 or 12 months) for precipitation (Bloomfield et al., 2015; Kumar et al., 2016). The approach of accumulating precipitation over a fixed period completely disregards spatial variability in the

response time of aquifers to precipitation (Stoelzle et al., 2014; Staudinger et al., 2015).

Laaha et al. (2016) suggest that "in absence of groundwater data, [streamflow] deficit volumes, representing the reduced outflow of stored sources in the catchment, may also be indicative for groundwater resources" (p.15). We agree that this approach might be more suitable as an indication of groundwater drought than a spatially-uniform SPI, but streamflow still integrates hydrological information over the whole catchment, disregarding spatial variability of groundwater levels within the

catchment. This is especially important in large catchments and regions with high spatial variability of landscapes and geology (e.g., Troch et al., 2009; Fan et al., 2013). Additionally, streamflow data are often not available in near real-time either, as pointed out by Laaha et al. (2016).

Others have explored the use of land surface models to quantify groundwater drought (Li and Rodell, 2015), which worked well on regional scales, but had much lower reliability on local scales. Like the approaches mentioned above, land surface

models can not capture the true spatial variability of groundwater droughts (Van Loon et al., 2012). Li and Rodell (2015) assimilated GRACE derived TWS anomaly in their land surface model and improved the results in terms of temporal correlation between modelled and observed groundwater drought, but because regionally averaged GRACE data were used the spatial representation of modelled groundwater droughts was not improved. This raises the questions whether direct use of GRACE fields would provide enough resolution to accurately characterise the spatial variability of groundwater droughts (AghaKouchak

et al., 2015). GRACE data has been used to quantify groundwater drought in a number of recent studies (Thomas et al., 2014; Breña Naranjo and Pedrozo Acuña, 2016), but their ability to provide better representation of the spatial variability of groundwater droughts remains to be tested.

Summarising, the major drawback of all approaches that are used as alternative for near real-time groundwater level data, is lack of spatial variability. There is an urgent need for a new approach that can account for spatial variability in groundwater

drought. Until groundwater level observations become widely available in near real-time, we have to find pragmatic solutions for quantifying groundwater drought during or shortly after a drought event. In this paper, we suggest to make use of known spatial variation in groundwater based on historic groundwater level datasets, assuming that heterogeneity of landscape and geology does not change over short timescales. If we know the spatial variation in the response time of the groundwater to precipitation from analysis of historical drought events, we can use that relationship to extrapolate to monitor a current drought.

These relationships have been determined for various parts of the world, e.g. Khan et al. (2008) in Australia, Mendicino et al. (2008) and Fiorillo and Guadagno (2012) in Italy, Bloomfield et al. (2015) in England, Li and Rodell (2015) in the USA, and Kumar et al. (2016) in the Netherlands and Germany. In this way, we combine readily available data (precipitation) with knowledge about the spatial variability of groundwater response derived from historic observations.

In this paper, we aim to analyse the 2015 groundwater drought in Europe by using a novel approach that exploits the pre-

determined historical relationships between precipitation and groundwater levels. The objectives of this study are: i) to test a





novel approach to obtain near real-time groundwater drought information for the 2015 European drought event, ii) to contrast the pattern of the 2015 groundwater drought with that of the 2003 groundwater drought, and iii) to discuss the differences between the groundwater drought and the meteorological drought (Ionita et al., 2016) and streamflow drought (Laaha et al., 2016). We focus on two regions in Europe, the eastern part of the Netherlands and southern Germany, for which groundwater

level data were available and historical relationships between precipitation and groundwater levels can be determined following the method of Kumar et al. (2016). Additionally, we explore the patterns of GRACE-TWS and groundwater anomalies for the 2003 and 2015 drought events. In the following section (Sect. 2), we describe the study areas and data. In Section 3, we explain the methodology of deriving historical relationships from standardised meteorological conditions and groundwater levels, and the GRACE data analysis. In Section 4 we show the 2015 groundwater drought for both regions and compare it with the 2003

groundwater drought. In Section 5 we discuss our methodology and findings, and compare them with the meteorological and streamflow drought analysed by Ionita et al. (2016) and Laaha et al. (2016). Concluding remarks are given in Sect. 6.

## 2   Study areas and data

This study focusses on two hydrogeologically different regions within Europe, i.e. the eastern Netherlands and southern Germany (Fig. 1 - upper panels). These regions were affected differently by the 2015 drought and are therefore suitable to study

spatial differences in groundwater drought using different methods. In our study, we used data of groundwater levels, meteorological variables (precipitation and temperature), and GRACE-TWS. For the meteorological datasets we used the free E-OBS data set (v13.0) from the European Climate Assessment & Dataset project (Haylock et al., 2008). These datasets are available at a $0.25° \times 0.25°$ spatial resolution and were created using the external drift kriging interpolation technique based on ground-based observation networks. We extracted daily precipitation totals and daily temperatures (average, maximum and minimum)

for the grid cells indicated in the green areas in Fig. 1. The daily precipitation totals were summed up to monthly precipitation totals. The temperature data were used to estimate daily potential evapotranspiration (PET) with the Hargreaves and Samani method, which only requires average, maximum and minimum daily temperature (Hargreaves and Samani, 1985). Other PET estimation methods could not be used, because of the limited availability of high-resolution meteorological variables over a long time period (1950-onwards). The daily PET values were then aggregated to monthly values.

We analysed monthly groundwater levels from a total number of 1991 and 49 observations wells located in southern Germany and the eastern Netherlands, respectively (Fig. 2 - upper left panels). The data for the German wells are acquired from the Bavarian Environment Agency (LfU Bayern) and the State Institute for Environment, Measurements and Nature Conservation Baden-Württemberg (LUBW), whereas the information for the Dutch wells were taken from the Dutch institute TNO (www. dinoloket.nl/). Most of these wells (90%) are located in shallow aquifer with an average depth to the water table within 20 m

below the ground surface. The length of records varied from well to well with a minimum of 10 years, starting from the year 1951 for the German wells and 1988 for the Dutch wells and ending in the year 2013. The selected wells are screened to have minimal anthropogenic influences that allow the understanding of the natural response of the groundwater to the precipitation





signal. Readers may refer to Kumar et al. (2016) for further details on information on the location, selection and/or processing of groundwater well records.

The seasonal and inter-annual variation of terrestrial water storage (TWS) is derived from the remotely sensed anomalies of Earth's gravity field retrieved by GRACE (version 5.0; Landerer and Swenson, 2012). The monthly anomalies are computed by removing the long-term mean estimated over the baseline period of Jan-2004 to Dec-2009 (NASA, 2016). The monthly GRACE derived TWS anomalies are available at $1° \times 1°$ spatial resolution from three different processing centres: the GeoForschungsZentrum (GFZ; Potsdam, Germany), the Center for Space Research at the University of Texas at Austin (United States), and the Jet Propulsion Laboratory (United States). We used the ensemble mean of these three GRACE-TWS products to reduce the noise (and scatter) among different TWS products. We used all monthly GRACE-TWS records that are available within the period 2002-2015.

## 3  Methodology

### 3.1  Analysis of standardised meteorological conditions and groundwater levels

In our novel approach, we analysed the spatio-temporal variability of the 2015 groundwater drought from the long-term relations between groundwater and driving meteorological conditions. For this purpose, the observations of monthly precipitation and groundwater levels were converted to the corresponding standardised anomalies (or probabilistic drought indices) denoted as Standardized Precipitation Evapotranspiration Index (SPEI; Vicente-Serrano et al., 2010) and standardised groundwater index (SGI; Bloomfield and Marchant, 2013). These indices represent standardised anomalies with respect to their own climatological estimates. SPEI accounts for both atmospheric water supply (precipitation) and demand (potential evapotranspiration; PET), which are both crucial variables for groundwater recharge, and therefore SPEI is assumed to correlate best to groundwater levels. We also considered the commonly used meteorological drought indicator Standardized Precipitation Index (SPI; McKee et al., 1993) that only takes into account precipitation.

Both meteorological drought indices (SPI and SPEI) can be estimated for different time scales by accumulating the corresponding monthly values (precipitation or precipitation minus PET) over different periods (e.g., 3, 6, 12, or 24 months) to reflect short and long-term (meteorological) droughts (e.g., Razei et al., 2013; Potop et al., 2014; Stagge et al., 2015; Barker et al., 2016). The SGI, on the other hand, is estimated at a monthly time scale (Bloomfield et al., 2015; Kumar et al., 2016) since groundwater often exhibits a high persistence (large memory), and therefore accounting for effects of the anomalous conditions during the preceding months.

Following Kumar et al. (2016), we used a non-parametric method to estimate the SGI, SPEI and SPI. Unlike fitting a predefined distribution function (e.g., Gamma function), the method uses a kernel density estimator to compute the cumulative probability distribution of the (accumulated) precipitation and groundwater data to estimate drought indices. The Gaussian kernel is selected here because of its unlimited support and the corresponding bandwidth is estimated via cross-validation approach (see Samaniego et al., 2013, for details). Furthermore, the bandwidth is computed for every calendar month and location separately to ensure comparability of resulting drought indices over time and space and to account for the high





seasonality exhibited by both variables. The computed indices are bounded between [0-1] with values below (above) 0.5 denote dry (wet) conditions. A drought is defined when the index value fall below a threshold of 0.2 or the 20th percentile, indicating an event with a re-occurrence period of five years. We note that the quantile-based indices are now increasingly used in drought studies (Sheffield et al., 2004; Andreadis et al., 2005; Vidal et al., 2010; Samaniego et al., 2013; Kumar et al.,

2016), and they can be easily transformed to the unbounded range of the standard normal distribution (Vidal et al., 2010). For example, the usual drought categories (see http://droughtmonitor.unl.edu) of moderate (e.g., SGI <= 0.2), severe (SGI <= 0.1), extreme (SGI <= 0.05) and exceptional (SGI <= 0.02) drought can easily be estimated.

We performed a regional scale groundwater drought analysis using indices computed at $0.25°$ grid resolution, at which the meteorological data are available. The SPI and SPEI were directly computed at this grid resolution for multiple accumulation

periods ranging between 1 and 48 months. Following Kumar et al. (2016), the gridded fields of SGI were estimated by first averaging the well-specific SGIs from all those wells that lie within the selected grid cell. And then, the averaged SGI values at each grid were converted into a percentile-based drought index using the non-parametric kernel density estimator approach. The number of wells within each $0.25°$ grid cell varied between 1 and 115 for Germany and between 1 and 6 for the Netherlands, with a median value of around 6 and 3 wells, respectively (Fig. 2 - upper left panels). The south German study region is covered

by 166 grid cells, whereas the Dutch part is covered by 17 cells.

We use a cross-correlation analysis to establish the linkage between meteorological and groundwater drought indices (i.e., SPI vs SGI and SPEI vs SGI) corresponding to different accumulation periods of SPI and SPEI (1-48 months). The goal is to identify the optimal accumulation period of meteorological based indices (SPI and SPEI) that can align with the SGI based groundwater droughts. The Spearman rank correlation coefficient (r) is used to quantify the strength of a monotonic

relationship between drought indices. This relationship is established at every grid cell separately; recognising that the optimal accumulation period varies from grid cell to grid cell, and a spatially fixed a-priori selection of the accumulation period (for computing the SPI and SPEI) would result in inadequate characterization of groundwater droughts (Kumar et al., 2016). The identified optimal accumulation periods of SPI and SPEI in the historical time period (1951-2013 for Germany and 1988-2013 for the Netherlands study regions) are then used to reconstruct the 2015 groundwater droughts over the study domain.

**3.2 Analysis of GRACE-TWS and groundwater anomalies**

GRACE-TWS shows a pronounced seasonal variation in western Europe that should be removed if we want to study drought in a similar way as with standardised indices such as SPEI and SGI. Because GRACE data is available for a relatively short period, we only applied a simple deseasonalisation or standardization, by estimating the z-score based on the monthly mean and standard deviation of the TWS-anomalies for each calender month and cell, separately.

GRACE-TWS represents variability in both groundwater and (near) surface water. We obtained (near) surface water storage (sum of soil moisture, canopy storage, and surface water) from four land surface models (VIC, Noah, MOSAIC, and CLM) available from the Global Land Data Assimilation System (GLDAS; Rodell et al., 2004). Monthly groundwater anomalies were then constructed after subtracting the GLDAS model derived (near) surface water storages from the TWS anomaly. To reduce the noise from individual GLDAS model outputs, we use the ensemble mean of the groundwater anomalies in our analysis.





## 4  Results

### 4.1  Characterisation of the 2003 and 2015 droughts in case study regions from precipitation, GRACE-TWS and groundwater observations

In this section we will characterise the 2003 and 2015 drought in our case study regions in the Netherlands and Germany from
5 readily available data, namely precipitation and GRACE-TWS. For 2003, we will also show the groundwater drought derived from borehole observations.

Average annual precipitation is relatively similar in the two case study regions (Fig. 1 - upper left panel), but drought can have a clearly distinct pattern. In 2003, both regions were relatively dry, whereas in 2015 southern Germany was the hotspot of the precipitation anomaly while the Netherlands had a relatively wet summer (Ionita et al., 2016; Laaha et al., 2016). This
difference is visible both in the spatial patterns and in the annual anomalies (Fig. 1 - upper and lower panels, respectively). Moreover, the southern Germany region has endured two consecutive precipitation deficits of more than 10% during 2014 and 2015, which may translate to a (larger) deficit in sub-surface water components (e.g., groundwater). In contrast, the region in the Netherlands endured two consecutive positive precipitation anomalies during the recent years. The two regions also exhibited a contrasting behaviour in terms of precipitation anomalies over the last decade: southern Germany experienced negative
anomalies (drier conditions), while the eastern Netherlands show positive anomalies (wetter conditions) in the majority of years during the last decade (Fig. 1 - lower panels).

Unfortunately, GRACE-TWS was not available for October and November 2015 due to technical problems with the satellites. Therefore, we mapped the months of August, September and December in Fig. 3, both for 2003 and 2015 and for both study regions. As evident from the figure, both years had a negative terrestrial water storage anomaly in August and September. The
20 TWS anomaly for the Netherlands was lower than that in Germany in both summers, ranging between -1 and -7 cm for the Netherlands and -6 and -14 cm for Germany. There was no apparent difference in the TWS spatial pattern between 2003 and 2015, with similar cells in the domain showing relatively higher or lower values. The only differences are the high TWS values in Scandinavia in 2015 and the drought recovery in December, which shows a southwest - northeast gradient in 2003 and a north - south pattern in 2015.

Figure 4 shows that after deseasonalisation, spatial patterns of TWS in the summers of 2003 and 2015 are still similar over the study regions. Again, only the recovery in December is clearly different between the two years. The timeseries for the grid boxes of Germany and the Netherlands (Fig. 4 - lower panels), do indicate different antecedent conditions in 2003 compared to 2015. Standardised TWS anomaly values were between 2 and 3 at the start of 2003 and between 0.5 and 1.5 in 2015, but this does not seem to have any significant effect on the drought severity, with standardised TWS anomalies for Germany around
30 -1.5 in both summers. The timeseries of TWS in Fig. 4 (grey lines in lower panels) also show that the spatial variability in standardised TWS anomaly values is limited, due to a relatively coarser resolution of GRACE dataset.

For 2003, the gridded SGI values derived from borehole observations show a pattern of high and low groundwater levels (Fig. 5). In Germany, the groundwater drought is clustered in the southeast, southwest and centre of the study area. In the Netherlands, the east and west are drier than normal and the centre of the study area is wetter than normal.





## 4.2 Relationship between standardised meteorological conditions and groundwater levels

The first step in quantifying the 2015 groundwater drought with our novel approach is determining the spatially explicit relationship between SGI and SPEI. Figure 2 (upper middle panels) shows that both in the Netherlands and Germany the optimal accumulation period of SPEI to match SGI varies substantially, between 1 and 48 months (4 years). The spatial pattern is

5 similar to that of Kumar et al. (2016), although the precipitation dataset and resolution were different (see Sect.5 Discussion for more details). In the Netherlands, we see relatively high SPEI accumulation periods in the centre and low accumulation periods in the west and east. In Germany, accumulation periods are relatively low in the south and high in the north, intermediate accumulation periods occur in the centre and high accumulation periods in the east and west.

As expected, the maximum correlation (the correlation obtained for the optimal SPEI accumulation period to match SGI)

did not show a specific spatial pattern (Fig. 2 - upper right panels). In the Netherlands, all correlations are above 0.6, while in Germany values below 0.6 occasionally occur, but never below 0.4. There is no relationship between the number of wells in the grid cell and the maximum correlation.

Since we calculated both SPI and SPEI for all grid cells in the study areas, we could compare the accumulation periods of SPI and maximum correlations between SPI and SGI with those of SPEI (Fig. 2 - lower panels). The clustering of points around

15 the 1:1-line illustrates that there is hardly any difference between using SPI and SPEI to determine the optimal accumulation period to match SGI. However, the Dutch grid cells tend to be all located above the 1:1-line indicating on average slightly higher accumulation periods and corresponding maximum correlations when using SPEI instead of SPI.

## 4.3 The 2015 groundwater drought derived from the relationship with meteorological conditions

From the relationship between SPEI and SGI for the historic period, we could calculate SGI values for the period for which we

did not have groundwater level observations (i.e. after 2013). This allowed us to map the SGI, derived based on the spatially varying optimally accumulated SPEI, during the drought event of 2015 (Fig. 6 and 7). The 2015 groundwater drought in Germany was extremely severe throughout the whole domain (i.e. SGI below the drought threshold of the 20th percentile for almost all grid cells in August, September and October 2015, and only a few grid cells in the Alps higher with values around 0.3 - 0.4; Fig. 6). The Netherlands, however, showed relatively high SGI values in 2015 (above the 50th percentile, so wetter

than normal, for all grid cells in the domain; Fig. 7).

When we compare the 2015 groundwater drought with the 2003 event, we notice that in both study regions there was a much larger spatial variability in SGI levels in 2003. In the Netherlands, 2003 actually showed up as a severe drought in some grid cells (Fig. 7 - upper panels), although on average the region was not classified as drought (Fig. 7 - lower panel). The reason for this is that some wells in the centre of the domain, those with optimal accumulation periods of more than 24 months (Fig. 2),

had higher than normal groundwater levels, even up to the 90th percentile. The same happened in Germany in 2003. Many grid cells in the northwest and east of the domain had above normal SGI (50th - 80th percentile; Fig. 6 - upper panels), while the grid cells in the centre, southwest and southeast had extremely low values. Although the values are slightly different, the SGI pattern derived from the SPEI and historic relationships between SGI and SPEI (Figs. 6 and 7 - upper panels) is very similar





to the one derived from borehole observations directly (Fig. 5). Despite the fact that the spatial average for the German region showed a drought in the summer of 2003 (below the 20th percentile, Fig. 6 - lower panel; in contrast to that of the Dutch region, Fig. 7 - lower panel), the SGI values ranged from 0 to around 0.95 (the 95th percentile), which is a very large spread. In the summer of 2015, the spatially averaged SGI was around 0.05 (5th percentile) for a few months and the range in SGI values

among grid cells was exceptionally low, i.e. in August 2015 only up to 0.3.

The crucial difference between the 2003 and 2015 summer droughts were the antecedent conditions. Since the majority of the groundwater responds to meteorological conditions (precipitation minus potential evaporation) accumulated over 6 to 24 months, with some exceptions going up to 48 months (Fig. 2), the relative wetness or dryness over the previous 6-48 months strongly influences the groundwater drought condition. Before 2003, the preceding three years had above average groundwater

levels, with spatially averaged SGI fluctuating around 0.7 (Fig. 7 - lower panel), culminating in a period of uniformly extremely high SGI values at the end of 2002 and start of 2003 (above 0.8). The antecedent conditions for 2015 show a number of years with below average SGI values and quite a severe drought in spring 2014. The timeseries of SGI in Germany also show that by the end of 2015 the groundwater had not recovered from the severe summer drought with spatially average values still around 0.05 (Fig. 7 - lower panel).

**4.4    The 2015 groundwater drought derived from GRACE**

The GRACE-TWS have been converted into groundwater anomalies by subtracting GLDAS model results of surface water stores. The resulting GRACE-GLDAS groundwater anomalies during the 2003 and 2015 drought events (Fig. 8, top panels) show a scattered pattern over Europe with our study area in the Netherlands showing wetter than normal groundwater storage during both droughts and our study area in Germany showing both wet and dry anomalies. The pattern of GRACE-GLDAS

groundwater anomalies does not correspond to patterns found in the gridded groundwater anomaly estimates from boreholes (Fig. 5) and does not show any similarity to other patterns of the 2003 and 2015 meteorological or hydrological drought (Ionita et al., 2016; Laaha et al., 2016) or to known differences in the aquifer characteristics or groundwater depth over Europe (Wendland et al., 2008; Fan et al., 2013).

The bottom two panels in Fig. 8 depict the monthly GRACE-GLDAS groundwater anomalies for every grid cell located

within the study domains (south Germany and central Netherlands) as well as their spatial averages. Except for a peak in 2014, the temporal variation in GRACE-GLDAS groundwater anomalies is limited. When compared to time series of groundwater anomalies from borehole observations (Fig. 5 in Kumar et al., 2016), no agreement is found. The removal of the seasonal cycle from the GRACE-GLDAS groundwater anomaly estimates (Fig. 9) slightly exaggerates the spatial pattern exhibited in Fig. 8 and also the time series do not change significantly. The individual GLDAS models show similar results (see Appendix A).

**5    Discussion**

The objectives of this study were: i) to test a novel approach to obtain near real-time groundwater drought information for the 2015 European drought event, ii) to contrast the pattern of the 2015 groundwater drought with that of the 2003 groundwater



drought, and iii) to discuss the differences between the groundwater drought and the meteorological drought (Ionita et al., 2016) and streamflow drought (Laaha et al., 2016).

## 5.1 Testing a novel approach to obtain near real-time groundwater drought information

The approach that we used to quantify the 2015 European groundwater drought in absence of near real-time groundwater level data is based on relationships between precipitation and observed historical groundwater levels. Our approach assumes that heterogeneity of landscape and geology does not change over the short timescales of the extrapolation (in this case 2013 to 2015) and that the relationship between standardised meteorological conditions (SPEI) and standardised groundwater levels (SGI) is relatively robust. The grid scale of 0.25° × 0.25° used for the analysis averages out sub-grid spatial variability, for example Kumar et al. (2016) showed a lower range in correlations between precipitation and groundwater from gridded data than from station data. However, the grid cells we used in this study are finer than those used in that previous study (0.25 instead of 0.5 degree spatial resolution) and the results still showed the expected spatial patterns (in the optimal accumulation period of SPI or SPEI). The big advantage of using the gridded E-OBS dataset in this study in contrast to local observation networks is that these meteorological data for Europe are freely available and are updated regularly.

The highly variable spatial pattern of the optimal accumulation period of SPEI again shows the importance of using a spatially-variable accumulation period to represent groundwater compared to using a spatially-uniform accumulation period. This confirms the results of Bloomfield et al. (2015) and Kumar et al. (2016) on the significance of the underlying land surface properties (geology) in transmitting the precipitation signal to groundwater levels.

The analysis using SPI instead of SPEI to represent meteorological conditions gave very similar results. This means that precipitation is the main driver of the optimal accumulation period of meteorological conditions to influence groundwater. This may be different in more arid regions where PET is a more important component in the water balance. For regions similar to the ones we analysed here, we expect that in absence of PET data SPI can be used instead of SPEI. The PET estimations used in this study are based on approximations of meteorological variables from average, maximum and minimum daily temperature. Temperature-based PET estimates in drought indices have been found to result in spurious trends in global drought (Sheffield et al., 2012). However, the temperature-based PET estimate used for estimation of SPEI is not expected to have influenced our results, because SPI showed similar accumulation periods as SPEI.

The GRACE-derived TWS anomaly data indicated drier than normal conditions in our case study regions. For groundwater drought analysis we needed to remove (near) surface water storage using a suite of four GLDAS models. Evaluation against gridded SGI values derived from well observations for the 2003 drought indicated that the spatial pattern of both the GRACE-TWS anomaly and GRACE-GLDAS groundwater anomalies did not show any agreement with observations. Therefore, the GRACE-GLDAS groundwater anomaly results should be used with caution and further research is needed to evaluate the reasons for the discrepancies between the GRACE-based model outcomes and groundwater observations. Other reasons for the unsuitability for GRACE-GLDAS groundwater anomaly results for real-time groundwater drought monitoring in these case study regions are: i) GRACE resolution may not be appropriate to understand the localized groundwater variability, and ii) model results are not always available in near real-time. From this analysis, we conclude that the GRACE-GLDAS groundwater





anomalies currently have large uncertainties that limit their application in real-time groundwater drought monitoring in this part of Europe.

## 5.2 Contrasting the pattern of the 2015 groundwater drought with that of the 2003 groundwater drought

In comparison to the benchmark 2003 drought event, the 2015 groundwater drought covered the entire German study domain, whereas the Netherlands were relatively wet. In 2003, both regions showed a patchy pattern with both extremely dry and (extremely) wet groundwater conditions. These differences in spatial pattern of the groundwater drought are related to antecedent conditions that affect different locations differently depending on the response time of the aquifer to meteorological conditions. In 2015, antecedent conditions were relatively dry over a long period (several years), so that both the quickly and slowly responding aquifers had low groundwater levels during the 2015 summer drought. However, in 2003, antecedent conditions were relatively dry over short timescales (a few months), causing quickly responding aquifers to show severe drought in summer 2003, but extremely wet over longer timescales (a year and longer), causing slowly responding aquifers to show wetter conditions in summer 2003. The reason for the extremely wet antecedent conditions in 2003 are the 2002 summer floods in Central Europe (e.g., Ulbrich et al., 2003a, b). These floods mainly affected our study region in Germany (Fig. 1 and 6), although also the Netherlands showed a relatively wet year in 2002 (Fig. 1 and 7). Without the 2002 floods, the groundwater drought of 2003 would have been more severe in southern Germany than that of 2015. This study also indicates that the recovery from drought in groundwater can be very patchy, with relatively fast recovery in locations with quickly responding aquifers (e.g. the eastern and western parts of the study area in the Netherlands in October 2003, Fig. 7, and the Alpine region in October 2015, Fig. 6) and slow recovery for locations with an aquifer response time of a few years (e.g. no recovery for many grid cells in Germany by the end of 2015, Fig. 6).

## 5.3 Discussing the differences between with meteorological and streamflow drought

In three previous papers, the 2015 European drought has been investigated from a meteorological and hydrological (i.e., streamflow) perspective and focussing on impacts (Ionita et al., 2016; Laaha et al., 2016; Van Lanen et al., 2016). The 2015 groundwater drought in Europe, as analysed in this paper, showed some striking similarities and differences with the meteorological and streamflow drought. Ionita et al. (2016) analysed 3-month accumulation periods for SPI and SPEI in June, July and August and concluded that an important difference between the 2003 and 2015 drought was that the winter and spring of 2015 were rather wet compared to dry preconditions in 2003. Laaha et al. (2016) indicated that also for streamflow drought no exceptionally dry conditions were found for the winter prior to the 2015 summer drought. Our analysis shows that for groundwater drought we need to look further back than the previous spring or winter, because accumulation periods of above 2 years are not uncommon and spatial variability in the memory of the groundwater to meteorological conditions is large.

Laaha et al. (2016) also found that the 2015 drought was less spatially extensive and more "patchy" in streamflow than that noticed in the 2003 drought. This was partly confirmed for groundwater with 2015 drought conditions apparent in Germany, but not in the Netherlands. However, the 2015 groundwater drought in Germany was much more spatially uniform than the 2003 groundwater drought because of much drier preconditions. Laaha et al. (2016) also mention the effect of the 2002 flood





event in Central Europe on the 2003 streamflow drought, but they only detect this effect in a shift of the onset of the drought. The reason for this could be that they used a fixed threshold to calculate streamflow drought, whereas standardised indices such as SGI are more comparable to a monthly-variable threshold (Van Loon, 2015).

Like streamflow drought, the severity of groundwater drought is determined by the combined effect of initial conditions (on different timescales), catchment functioning and aquifer characteristics, and atmospheric conditions (Laaha et al., 2016). In 2015, the combination of these factors resulted in a uniformly severe groundwater drought in Germany, which lead to socio-economic impacts in the region, e.g. water shortage for cattle due to dried boreholes (Van Lanen et al., 2016). These impacts are expected to be less severe in more patchy groundwater drought events, like the 2003 event, because in these cases other boreholes in the region would still exhibit above normal groundwater levels.

## 6 Conclusions

In this research we proposed an approach to monitor groundwater drought for the 2015 European drought event as alternative to real-time groundwater observations. This alternative method, based on extrapolation of the pre-determined relationship between meteorological conditions and observed groundwater levels, seemed suitable, as it reproduced the 2003 groundwater drought in both case study regions (derived from borehole observations) and showed expected patterns of spatial variability for 2015, related to aquifer characteristics, antecedent conditions and meteorological drought. This novel approach performed much better than the satellite-based GRACE-TWS and GRACE-GLDAS groundwater anomalies. GRACE-TWS has a too coarse resolution to represent the observed spatial variability in groundwater drought and GRACE-GLDAS models were not able to simulate groundwater anomalies realistically for our study areas. We found a high spatial variability in the optimal accumulation period of SPEI to represent anomalies in groundwater levels (SGI), which had significant influence on the spatial pattern of the 2003 and 2015 groundwater drought. These results signify that a spatially-uniform accumulation period should not be used to represent hydrological drought over larger areas. The analysis of both SPI (representing accumulated precipitation anomalies) and SPEI (representing accumulated anomalies in precipitation minus potential evaporation) showed similar results, indicating that precipitation is the main driver of the optimal accumulation period of meteorological conditions to influence groundwater in our study areas.

Our analysis shows that the 2015 groundwater drought was uniformly severe over southern Germany due to a combination of a relatively dry summer and dry preceding seasons and years. This is in contrast with the 2003 groundwater drought, which showed a severe drought in areas with quickly responding aquifer systems, both in the Netherlands and in Germany, in response to a dry summer, but wetter than normal conditions in areas with slowly responding aquifer systems, in response to a wet preceding two years. Especially in Germany, the 2002 summer floods still influenced some groundwater levels in the summer of 2003.

Groundwater drought is notably different from meteorological drought and streamflow drought. For example, the meteorological drought of 2015 was actually found to have originated much later than that of 2003, with persistent anticyclonic circulation starting at the end of spring and the end of winter for 2015 and 2003, respectively (Ionita et al., 2016). The stream-



flow droughts, which respond relatively quickly to precipitation deficits, matched up quite well with these meteorological droughts, with the 2015 streamflow drought having in general shorter durations than the 2003 streamflow drought (Laaha et al., 2016). However, (spatial) differences in the severity of streamflow drought within the larger European study area were apparent and were found to be related to differences in catchment functioning that determine how far back antecedent conditions influence streamflow. This effect is even stronger in groundwater: in many areas extending back to the previous winter is not enough to adequately capture the groundwater response to precipitation signals.

We recognise that our approach of using the pre-determined relationship between meteorological conditions and observed groundwater levels is crude and has uncertainties. It does, however, provide a first-order look into the spatio-temporal patterns of current and recent groundwater droughts based on meteorological indices. This is crucial for pro-active drought management in absence of real-time groundwater observations. With this work, however, we also want to promote more long-term groundwater measurement and international sharing of groundwater level data.

## 7 Code availability

The code for calculating the standardised indices and gridded estimates of groundwater drought are available from the authors. Please contact Rohini Kumar at rohini.kumar@ufz.de.

## 8 Data availability

The data used in this study are freely available from different sources. The Dutch groundwater level data are available online from the Dutch institute TNO (www.dinoloket.nl/). The E-OBS dataset is available from the EU-FP6 project ENSEMBLES and the European Climate Assessment & Dataset project (http://ensembles-eu.metoffice.com and http://www.ecad.eu). GRACE-TWS and GRACE-GLDAS data are free to download from http://grace.jpl.nasa.gov/. The German groundwater level data not publicly available, because they have been collected by the Bavarian Environment Agency (LfU Bayern) and the State Institute for Environment, Measurements and Nature Conservation Baden-Württemberg (LUBW) for use in GLOWA-Danube project (www.glowa-danube.de), funded by the German Ministry of Education and Research (BMBF). These institutes have restricted further distribution of the data.

## Appendix A: GRACE-GLDAS groundwater anomalies simulated by four different models

To test the effect of using GLDAS model simulations, we used the CLM, NOAH, VIC and MOSIAC models to remove the total (near) surface water storage anomaly (TOW = Soil moisture + Canopy Storage + Snow water equivalent) from the GRACE-TWS anomaly. We calculated the ensemble mean of the resulting GRACE-GLDAS groundwater anomalies of the four models, but also plotted at the four models separately (Figs. A1 - A8). None of the models was able to reproduce the observed groundwater drought pattern for 2003 (Fig. 5) or the expected pattern for 2015 (Figs. 6 and 7).



*Author contributions.* A.F. Van Loon and R. Kumar designed the study, V. Mishra provided GRACE data, R. Kumar performed the analyses with feedback from A.F. Van Loon and V. Mishra. A.F. Van Loon prepared the manuscript with contributions from all co-authors.

*Competing interests.* The authors declare that they have no conflict of interest.

*Acknowledgements.* We would like to thank the Bavarian Environment Agency (LfU Bayern) and the State Institute for Environment,
5 Measurements and Nature Conservation Baden-Württemberg (LUBW) who provided the southern Germany groundwater data sets for use in GLOWA-Danube (www.glowa-danube.de), a project funded by the German Ministry of Education and Research (BMBF). We also thank the Dutch institute TNO for making the Dutch groundwater level data available (www.dinoloket.nl/). We acknowledge the E-OBS dataset from the EU-FP6 project ENSEMBLES (http://ensembles-eu.metoffice.com) and the data providers in the European Climate Assessment & Dataset project (http://www.ecad.eu). We thank NASA and the GLDAS model groups for providing the GRACE-TWS and GRACE-GLDAS
10 data (http://grace.jpl.nasa.gov/). We also want to acknowledge A. Akarsh for his help in GRACE and GLDAS data processing. This research is partly funded by NWO Rubicon project no. 2004/08338/ALW. This paper was developed within the framework of the UNESCO-IHP VIII FRIEND programme (EURO-FRIEND - Low flow and Drought group) and supports the Helmholtz Climate Initiative REKLIM project.





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







**Figure 1.** Long-term mean annual precipitation (1950-2015) and precipitation anomalies in the years 2003 and 2015 (% difference from long-term mean) for our study domain in Europe. The green boxes indicate the study regions and the black dots represent the grid cells used for the analysis. The bottom two panels show the annual precipitation anomalies (% deficit from the long term mean) for the two study regions located in southern Germany and the eastern Netherlands.





**Figure 2.** The location of groundwater wells and number of groundwater wells in every $0.25°$ grid cell, the optimal accumulation $A$ (month) and the maximum correlation $r_{\mathrm{m}}$(-) between the gridded SGI and SPEI for German (top panel) and Dutch (middle panel) data sets. The bottom panels show the correspondence of $A$ and $r_{\mathrm{m}}$ estimated from (SPI vs. SGI) and (SPEI vs. SGI).




**Figure 3.** The GRACE derived Terrestrial Water Storage (TWS) anomalies (cm) for the available three common months (August, September, and December) during the 2003 and 2015 drought events (top panel). The bottom two panels depict the monthly TWS anomalies for every grid cell located within the study domains (shown in the green boxes in the top panels; south Germany and central Netherlands) as well as their spatial averages.

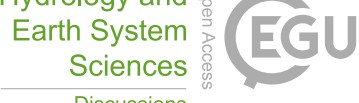

**Figure 4.** Same as Fig. 3, but after standardisation of the TWS anomalies to de-emphasise the seasonality component of TWS anomalies.



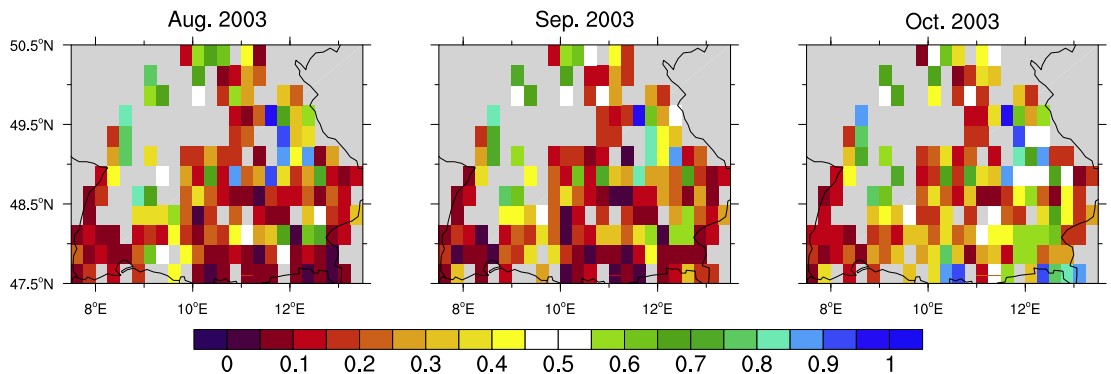

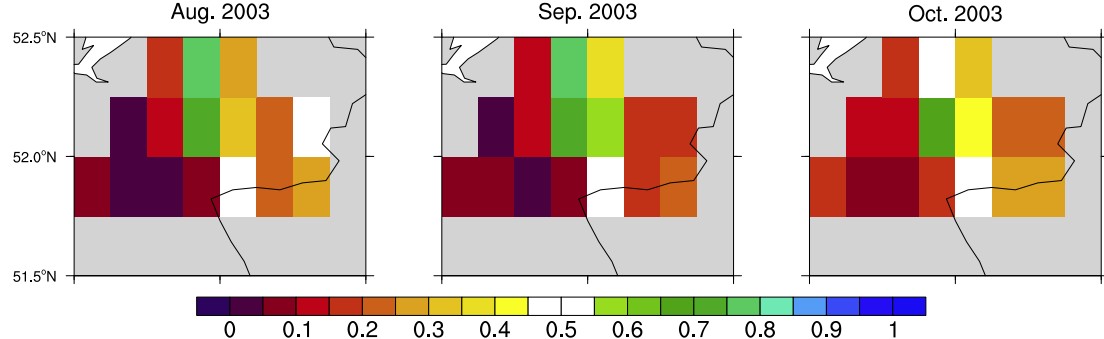

**Figure 5.** The groundwater drought (SGI) during the months of August, September, and October for the 2003 drought event over the south Germany and east Netherlands study domains, derived from borehole observations and recalculated to 0.25° grid cells for comparability.





**Figure 6.** The reconstructed groundwater drought (SGI) based on the SPEI with spatially varying (optimal) accumulation periods during the months of August, September, and October for the 2003 and 2015 drought event over the south Germany study domain. The monthly time series of the optimally accumulated SPEI for every 0.25° grid cell and the respective spatial averages (dark red) are depicted in the bottom panel. The orange dashed line depicts the drought threshold ($\tau$) of 0.2.







**Figure 7.** Same as Fig. 6, but for the central Netherlands study domain.



**Figure 8.** Groundwater anomalies (cm) based on GRACE and GLDAS model outputs for the available three common months (August, September, and December) during the 2003 and 2015 drought events (top panel), estimated by removing the total surface water storage anomaly (TOW = Soil moisture + Canopy Storage + Snow water equivalent) from the GRACE-TWS anomaly (Fig. 3). The TOW estimates are based on mean simulations from four GLDAS models (CLM, NOAH, VIC and MOSIAC). The plots are based on the ensemble mean of these four model outputs. The bottom two panels depict the monthly GRACE-GLDAS groundwater anomalies for every grid cell located within the study domains (shown in the green boxes in the top panels; south Germany and central Netherlands) as well as their spatial averages.



**Figure 9.** Same as Fig. 8, but after standardisation of the GRACE-GLDAS groundwater anomalies to de-emphasise the seasonality component of the anomalies.





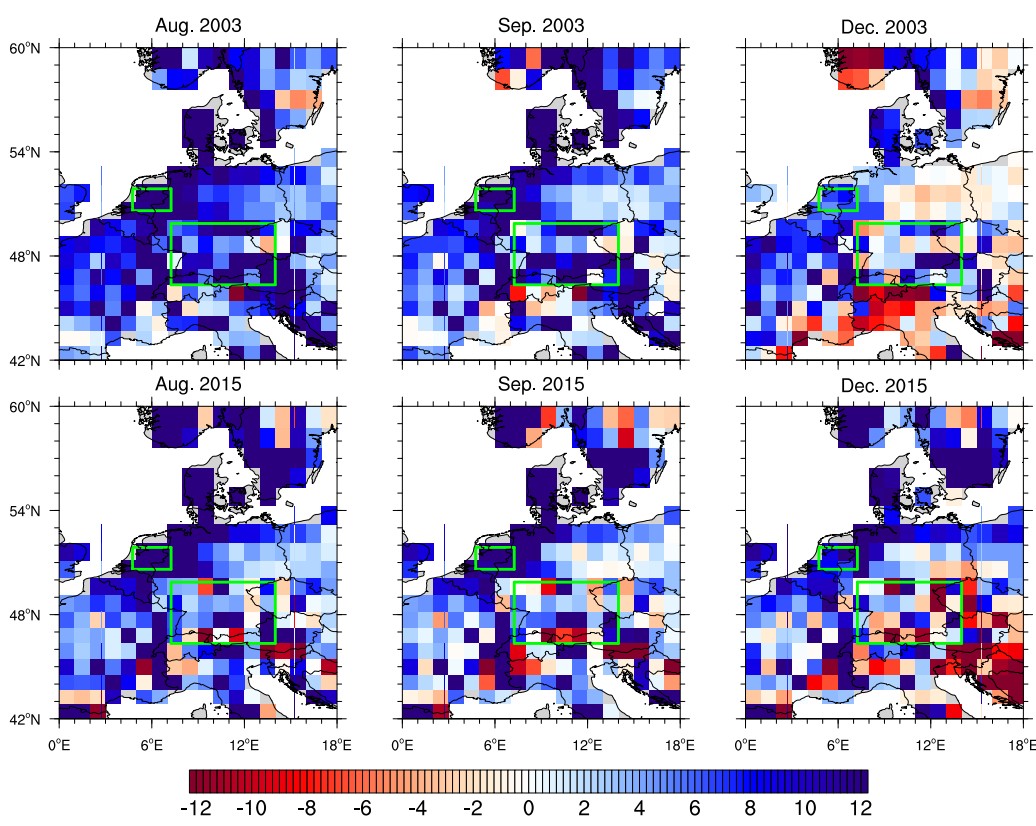

**Figure A1.** Groundwater anomalies (cm) based on GRACE and VIC model outputs for the available three common months (August, September, and December) during the 2003 and 2015 drought events (top panel), estimated by removing the total surface water storage anomaly (TOW = Soil moisture + Canopy Storage + Snow water equivalent) from the GRACE-TWS anomaly (Fig. 3).





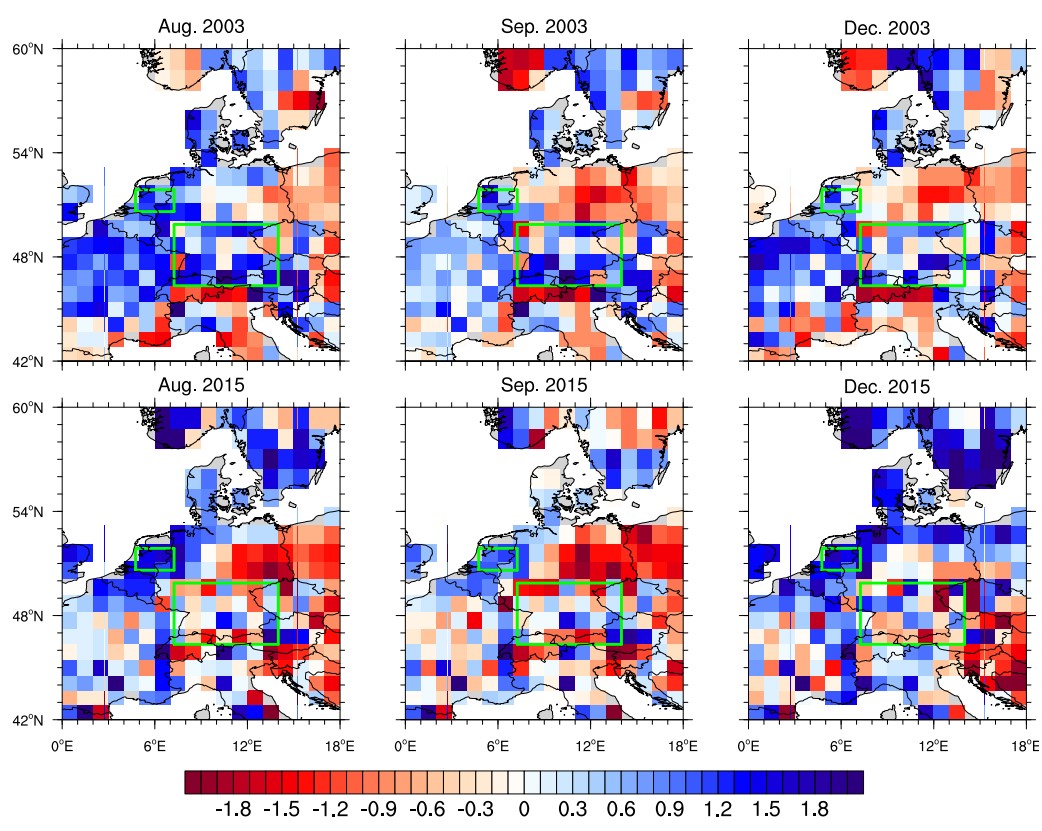

**Figure A2.** Same as Fig. A1, but after standardisation of the GRACE-VIC groundwater anomalies to de-emphasise the seasonality component of the anomalies.



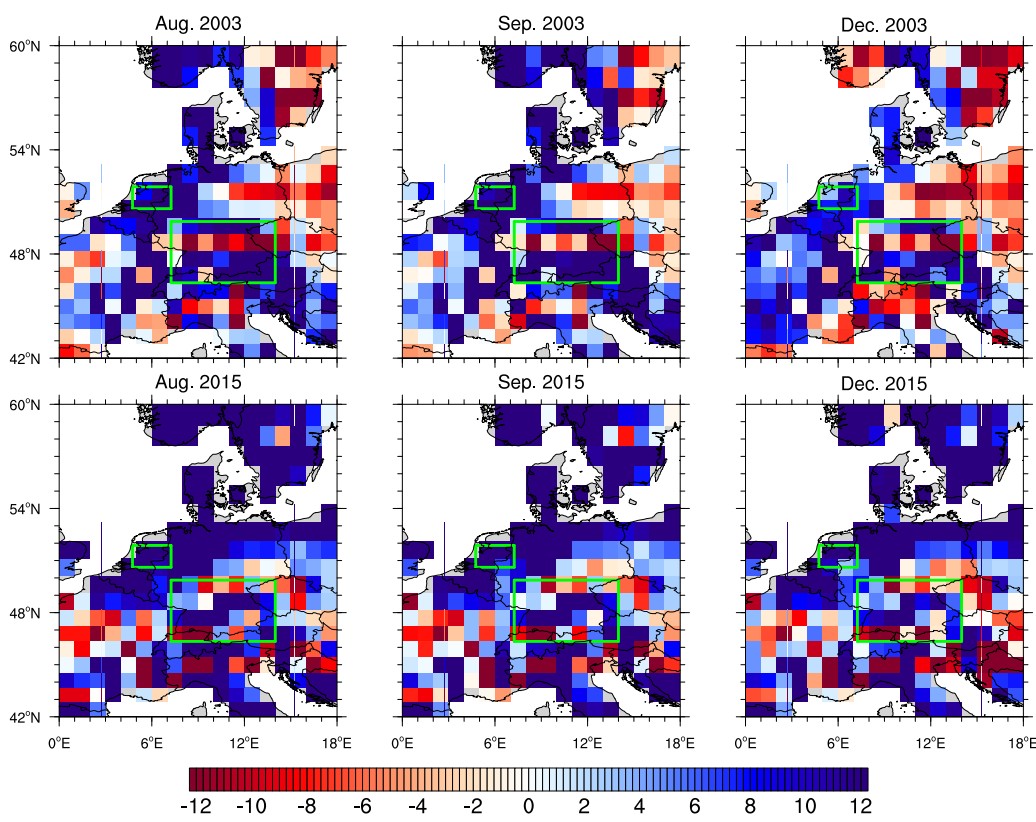

**Figure A3.** Groundwater anomalies (cm) based on GRACE and MOSAIC model outputs for the available three common months (August, September, and December) during the 2003 and 2015 drought events (top panel), estimated by removing the total surface water storage anomaly (TOW = Soil moisture + Canopy Storage + Snow water equivalent) from the GRACE-TWS anomaly (Fig. 3).





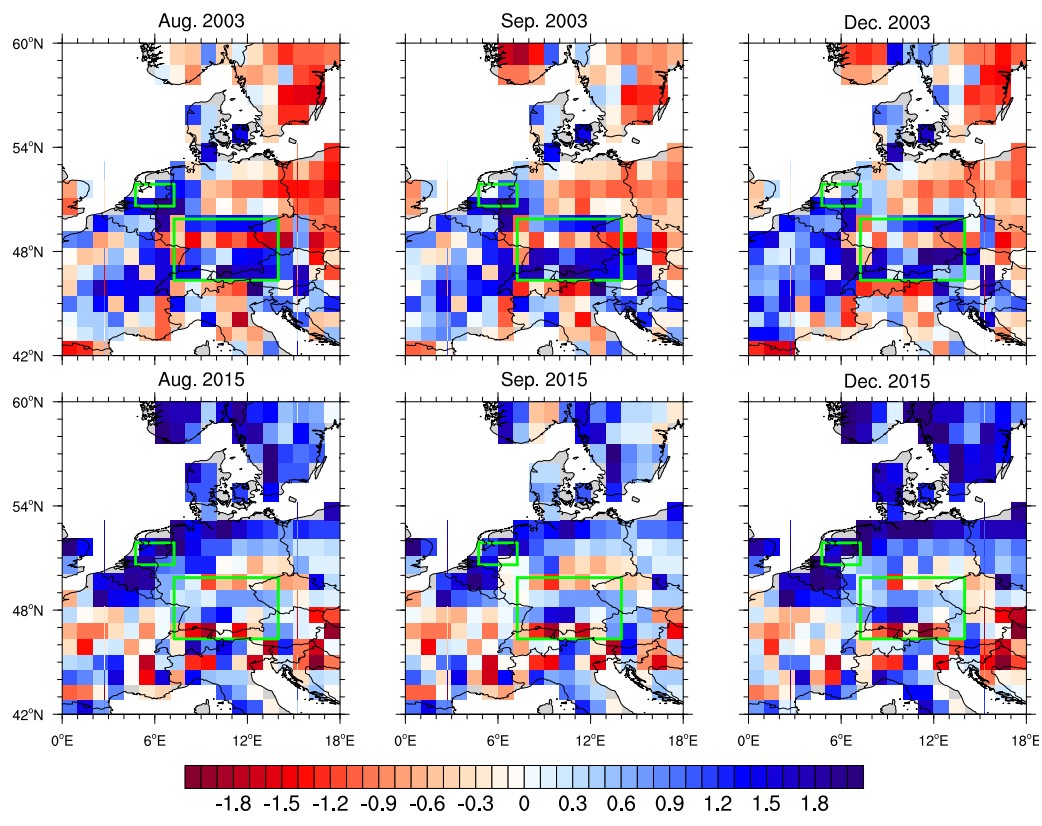

**Figure A4.** Same as Fig. A1, but after standardisation of the GRACE-MOSAIC groundwater anomalies to de-emphasise the seasonality component of the anomalies.





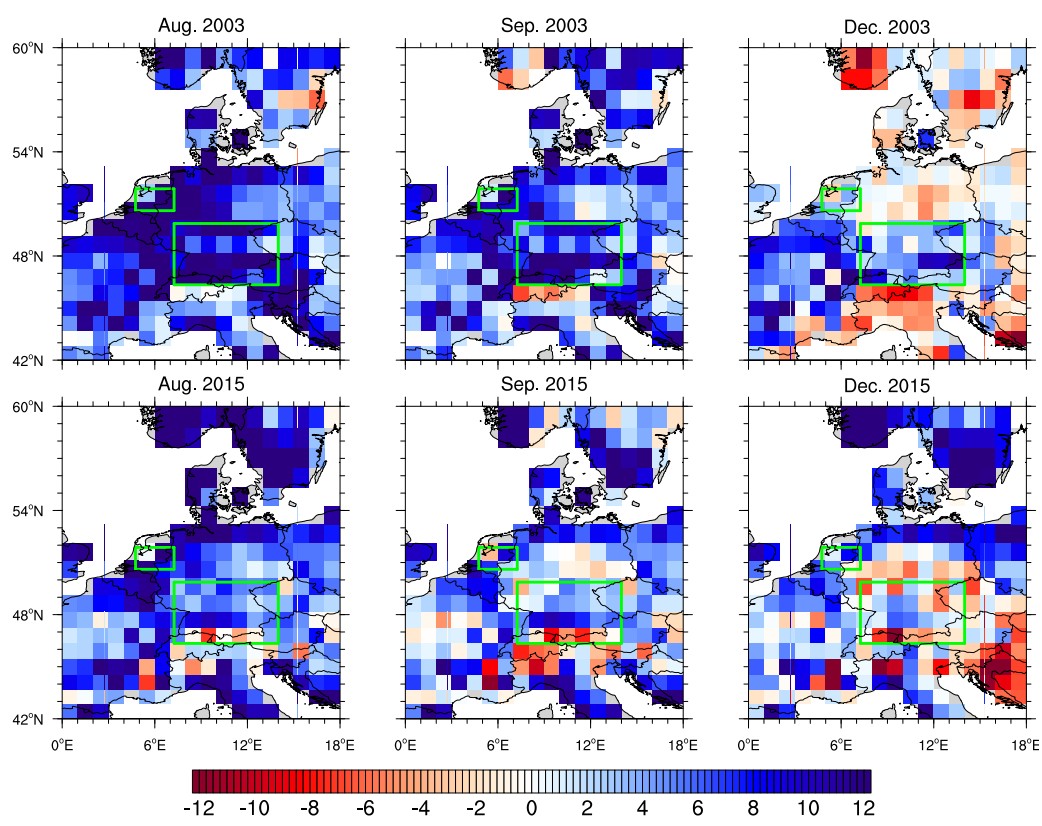

**Figure A5.** Groundwater anomalies (cm) based on GRACE and NOAH model outputs for the available three common months (August, September, and December) during the 2003 and 2015 drought events (top panel), estimated by removing the total surface water storage anomaly (TOW = Soil moisture + Canopy Storage + Snow water equivalent) from the GRACE-TWS anomaly (Fig. 3).





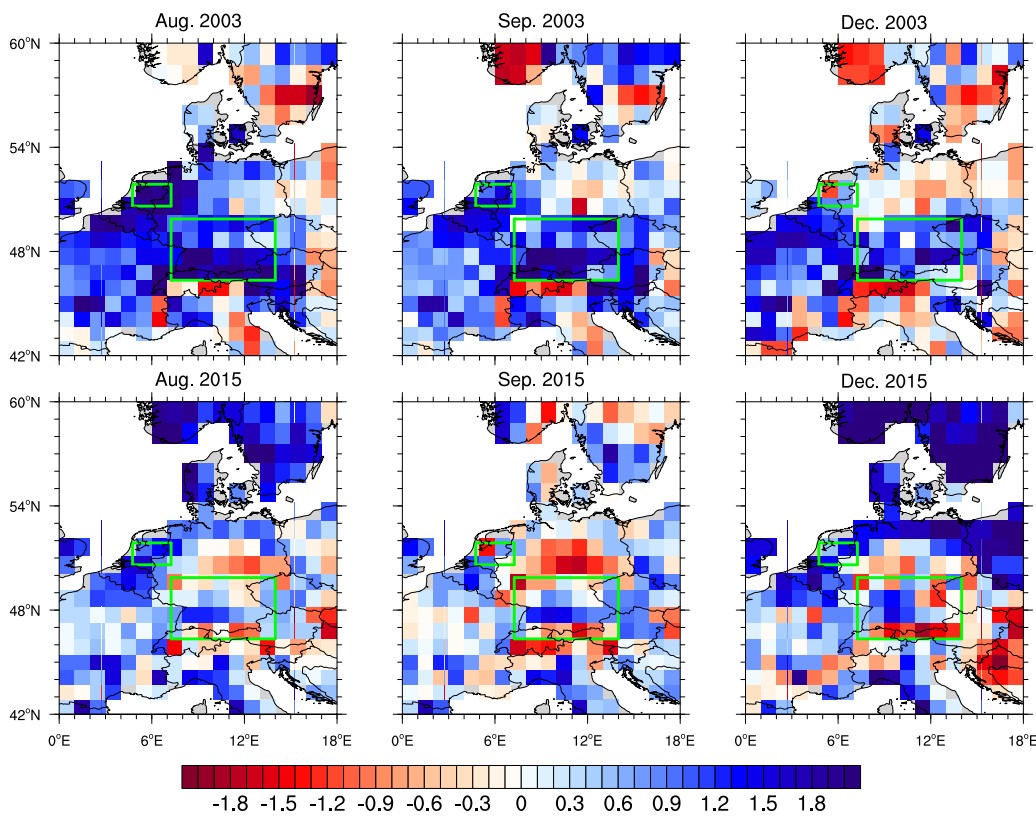

**Figure A6.** Same as Fig. A1, but after standardisation of the GRACE-NOAH groundwater anomalies to de-emphasise the seasonality component of the anomalies.





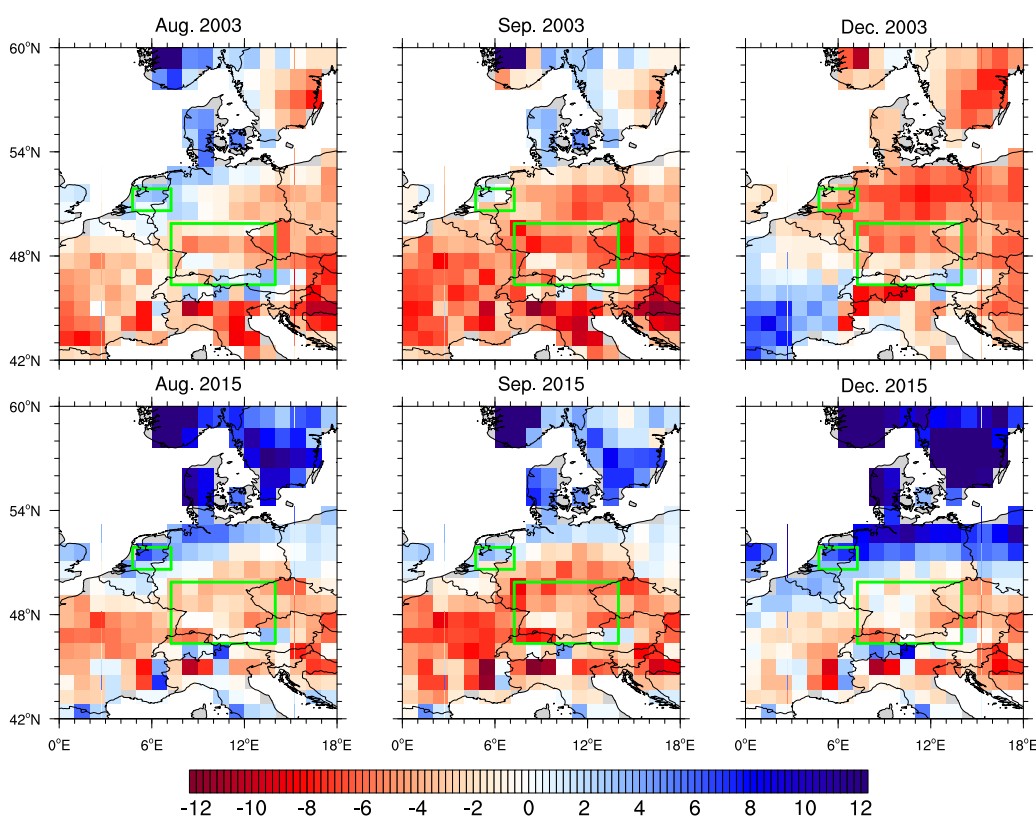

**Figure A7.** Groundwater anomalies (cm) based on GRACE and CLM model outputs for the available three common months (August, September, and December) during the 2003 and 2015 drought events (top panel), estimated by removing the total surface water storage anomaly (TOW = Soil moisture + Canopy Storage + Snow water equivalent) from the GRACE-TWS anomaly (Fig. 3).





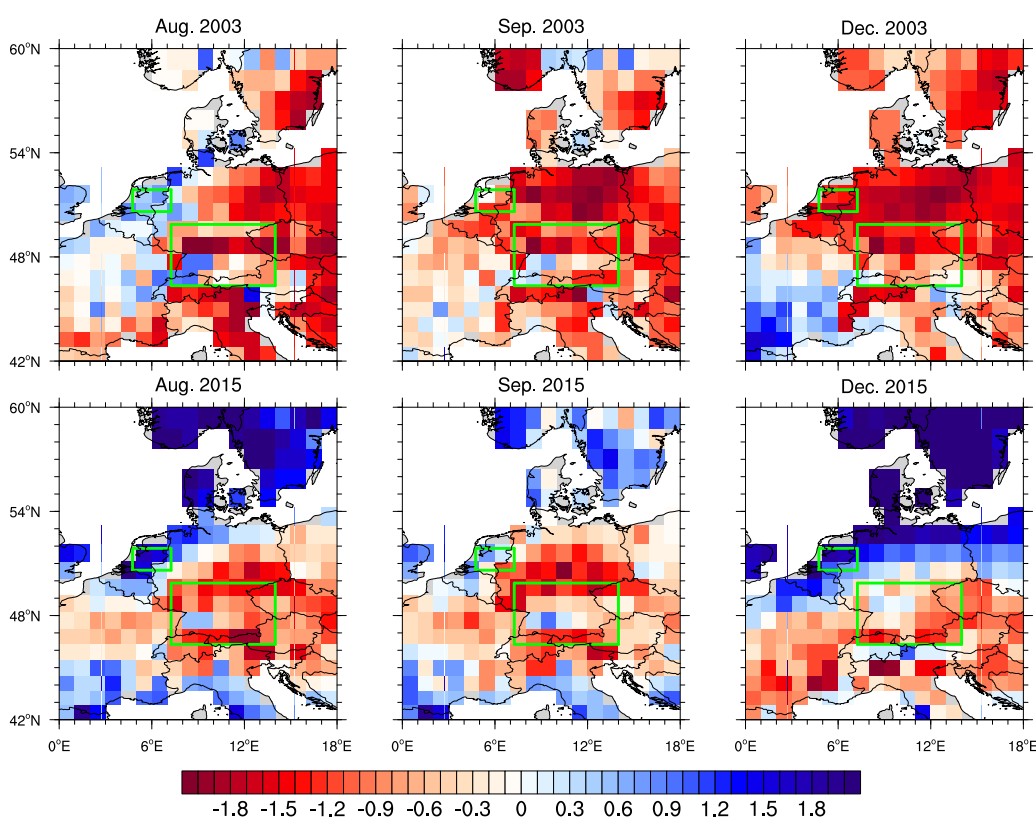

**Figure A8.** Same as Fig. A1, but after standardisation of the GRACE-CLM groundwater anomalies to de-emphasise the seasonality component of the anomalies.