# Peer review of "Testing the use of standardised indices and GRACE satellite data to estimate the European 2015 groundwater drought in near real-time"

_Hydrology and Earth System Sciences, 2016_

## Referee Comment (RC1) · M. Fendekova (Referee) · 29 Dec 2016

General comments The problem of groundwater level drought evaluation is still less often studied in comparison to meteorological or streamflow droughts. Evaluation of groundwater drought is complicated because of many factors influencing its development and persistence even within relatively small areas. Therefore the paper of authors van Loon et al. is highly appreciated. The paper is well structured, based on the present-day knowledge of the meteorological and hydrological drought evaluation, and techniques of groundwater drought assessment. The input data, methods and results are well described. The discussion and conclusions are clear, well understandable. The concept of accumulation period estimation and its relation to SPI and SPEI

indices led to adequate results within two different areas (southern Germany, Netherlands). Use of satellite-based GRACE-TWS and GRACE-GLDAS showed that because of coarse resolution the models were not able to simulate groundwater anomalies realistically.

Scientific questions/comments: 1/ Did the authors think about looking at other groundwater parameters, as baseflow or spring yield from the point of drought occurrence? 2/ Hydrogeological conditions and recharge-discharge relationships of an aquifer are more purely represented by spring yields and their changes during the meteorological drought periods. Therefore, maybe the next step in groundwater drought research should be the spring yields drought study. However, in some hydrogeological conditions (as in the Netherlands) the data availability might be very limited. 3/ Do the authors recommend the use of SPI index which calculation is easier than the SPEI index giving the comparable good results?

Technical comments: Despite of the reference to Kumar et al. (2016) I would appreciate at least the very short description of major differences in hydrogeological conditions of both areas used in the study. There are not comments to the English language which is excellent, and to figures quality.

---

## Referee Comment (RC2) · J.P. Bloomfield (Referee) · 4 Jan 2017

General comments: The paper was enjoyable to read, is clearly written and addresses an important set of related research questions on the topic of the predictability of heterogeneous groundwater systems to drought under conditions of incomplete information. My general comments on the paper relate to two points: a.) the framing of the paper, i.e. need for near real-time groundwater drought data and the ability to develop calibrated modelling systems in advance of groundwater droughts, and b.) the title of the paper. As an aside, the authors should be congratulated on seeking to publish a negative finding (their observation that, with regard to the present study, GRACE –TWS did not appear to be suitable for use in real-time groundwater drought assessment) –

such findings can be very useful, but are often under-reported.

The need for near real-time groundwater drought data. The paper explores two possible approaches to estimating, in near real-time, spatially varying groundwater levels under conditions of drought by: i.) establishing correlations between a standardised hydrological index (in this case SPEI, although SPI "showed similar results") and observed groundwater levels, and ii.) estimating groundwater anomalies by subtracting modelled surface water stores from GRACE-TWS data (that represents variability in both groundwater and near surface water). Once calibrated or established such approaches could in theory inform near real-time decisions on groundwater management during future droughts in the absence of groundwater level observations.

In this context, the authors note in their conclusions the following: "We recognise that our approach of using the pre-determined relationship between meteorological conditions and observed groundwater levels is crude and has uncertainties. It does, however, provide a first-order look into the spatio-temporal patterns of current and recent groundwater droughts based on meteorological indices". An alternative approach to modelling the spatial variability of groundwater droughts in near real-time using the available a priori meteorological and groundwater level data could have been developed. Simple lumped parameter models are increasingly used to model groundwater levels; make almost no assumptions about hydrogeological setting; and, if there are sufficient sites, capture and reflect spatially varying responses of groundwater systems. Lumped parameter groundwater models are already used for operational hydrological services, such as UK Hydrological Outlooks http://www.hydoutuk.net/methods/groundwater/. In addition it is possible to constrain the uncertainty in such models as well as the success of their predictions. For example, see work using such models from the UK (Mackay et al., 2014; 2015; Jackson et al., 2016; Marchant et al., 2016). The paper would have a more rounded context if the introduction includes a discussion of such lumped parameter models and the pros and cons of the adopted approach compared with a lumped parameter modelling approach.

It should be noted that the SPEI/SGI correlation approach described in the paper produces a calibrated but un-validated correlation and that any SGI values modelled using SPEI driving data will have unconstrained uncertainties. Whereas, using the same calibration and driving data, if multiple lumped parameter groundwater level models are produced they can be both calibrated and validated and uncertainties estimated for each of the predictions of groundwater levels. The main 'cost' in this latter case would be the time involved in producing multiple individual calibrated lumped-parameter models although the process can to some extent be automated.

The authors also note in their conclusion: "With this work, however, we also want to promote more long-term groundwater measurement and international sharing of groundwater level data". I entirely agree with this statement. For example, throughout Europe groundwater levels and spring discharges are extensively monitored by a wide range of organisations and institutions for a variety of purposes. Some of this information is freely available on the web, however, much of it is not readily available and certainly not in near real-time. Significant advances in the effective management of groundwater resources during droughts could be achieved with better co-ordination and sharing of groundwater data at the European scale. Such a freeing-up of information would in one step obviate the need to model 'near real-time' groundwater levels as described in the current paper and would enable more effective modelling of 'near future' groundwater levels using more sophisticated lumped parameter-type models.

Title of the paper. The paper aims to establish an approach for estimating near real-time groundwater levels during episodes of groundwater drought in the absence of groundwater observations. It uses the expression of the European drought of 2015 from two regions, in Germany and the Netherlands, to test two alternative modelling approaches. However, I don't feel that it is appropriate to suggest that it provides a coherent insight into the groundwater aspects of the European of drought 2015. Consequently, I'd suggest an alternative title such as: "Estimation of near real-time groundwater drought status in the absence of observed groundwater level data".
Specific comments: P3., last para – given my comments above regarding the title of the paper, I don't think that the statement "In this paper, we aim to analyses the 2015 groundwater drought in Europe . . ." is quite right. I suggest re-phrasing to something like "In this paper, we asses two alternative approaches to model near real-time groundwater drought . . ."

P4., last para - when working with standardised indices such as SPI or SGI it is common practice to produce standardised values on a common time period and with a minimum record length of 30 years (McKee et al 1993 and others). What errors have been introduced into the analysis due to differences in record lengths within and between the two study regions and do these errors effect the conclusions of the study given that "The length of [the groundwater level] records varied from well to well with a minimum of 10 years, starting from the year 1951 for the German wells and 1988 for the Dutch wells and ending in the year 2013"?

P5-6. Section 3.1 and Fig 2 (top and middle panels) – the SGI data is the same as Kumar et al. (2016) and the difference between Fig 2 of Kumar et al. (2016) and Fig 2 (top and middle panels) of this study is that the latter uses a more refined grid for the analysis. What are the implications, if any, of the reduced number of groundwater level time series observations within the smaller grid cells of the present study on the averaging procedure to obtain a representative SGI for each cell?

P6., last para – "To reduce the noise from individual GLDAS model outputs, we use the ensemble mean of the groundwater anomalies in our analysis". It would be nice to have a bit more information on the scale and nature of the noise in the GLDAS model outputs, perhaps scaled as a function of the GRACE-TWS data? Is there any temporal or spatial structure in the noise relevant to the two study areas and the periods of calibration and modelling?

P8, para 2 - Once an optimal accumulation period has been established for each cell, why has the maximum (point?) correlation between pairs SPEI/SGI of time series been

plotted in Fig 2, wouldn't a representative or (grid) average correlation corresponding to the optimal accumulation period be more instructive than the maximum correlation? It would of course be likely to be lower than the reported correlations.

P10., para 1 of Section 5.1 – as above, I suggest re-drafting to the first sentence to "We assessed two alternative approaches to model groundwater drought in the absence of . . . . . ."

P10., para 2 of Section 5.1 – It is stated that "The analysis using SPI instead of SPEI to represent meteorological conditions gave very similar results. This means that precipitation is the main driver of the optimal accumulation period of meteorological conditions to influence groundwater. This may be different in more arid regions where PET is a more important component in the water balance. For regions similar to the ones we analysed here, we expect that in absence of PET data SPI can be used instead of SPEI". Although not critical to the paper, this is an interesting observation, but I'm not sure that the interpretation is correct. Bloomfield and Marchant (2013) demonstrated that the optimal accumulation period for SPI/SGI correlation scaled as a function of the autocorrelation range of the groundwater level time series (mmax) (Bloomfield and Marchant, 2013, Fig.10), which in turn was shown to be a function of unsaturated zone thickness and log-hydraulic diffusivity of the aquifer (Bloomfield and Marchant, 2013, Fig. 13), i.e. that it was necessary to invoke aquifer and catchment processes responsible for attenuation of meteorological signals to explain the optimal accumulation period. Assuming that similar relationships hold for SPEI/SGI then I don't think that precipitation is the main control on the optimal accumulation period as stated, rather it is catchment and aquifer characteristics. PET would be expected to have a very limited effect on groundwater levels once a drought has been established when the main cause of groundwater decline would be natural groundwater recession due to groundwater discharge in the absence of precipitation. Note that under drought conditions soil moisture deficits are likely to be very high so limiting the effect of PET.

P12., end of first para of section 6 – again the statement that "The analysis of both
[Figure]

SPI (representing accumulated precipitation anomalies) and SPEI (representing accumulated anomalies in precipitation minus potential evaporation) showed similar results, indicating that precipitation is the main driver of the optimal accumulation period of meteorological conditions to influence groundwater in our study areas". See my comments above. I don't think that this interpretation is correct and that the optimal accumulation period is a function of catchment and aquifer characteristics not precipitation.

References: Bloomfield, J.P.; Marchant, B.P. 2013 Analysis of groundwater drought building on the standardised precipitation index approach. Hydrology and Earth System Sciences, 17. 4769-4787. 10.5194/hess-17-4769-2013

Bloomfield, J.P., Marchant, B.P., Bricker, S.H., and Morgan, R.B. 2015. Regional analysis of groundwater droughts using hydrograph classification. Hydrology and Earth System Sciences, 19 (10). 4327-4344. 10.5194/hess-19-4327-2015

Kumar, R., Musuuza, J. L., Van Loon, A. F., Teuling, A. J., Barthel, R., Ten Broek, J., Mai, J., Samaniego, L., and Attinger, S. 2016. Multiscale evaluation of the Standardized Precipitation Index as a groundwater drought indicator, Hydrology and Earth System Sciences, 20, 1117–1131, doi:10.5194/hess-20-1117-2016, http://www.hydrol-earth-syst-sci.net/20/1117/2016/.

Mackay, J.D., Jackson, C.R., Brookshaw, A., Scaife, A.A., Cook, J., and Ward, R.S. 2015. Seasonal forecasting of groundwater levels in principal aquifers of the United Kingdom. Journal of Hydrology, 530. 815-828. 10.1016/j.jhydrol.2015.10.018

Mackay, J.D., Jackson, C.R., and Wang, L. 2014. A lumped conceptual model to simulate groundwater level time-series. Environmental Modelling and Software, 61. 229-245. 10.1016/j.envsoft.2014.06.003

Marchant, B., Mackay, J., and Bloomfield, J.P. 2016. Quantifying uncertainty in predictions of groundwater levels using formal likelihood methods. Journal of Hydrology, 540. 699-711. 10.1016/j.jhydrol.2016.06.014

McKee, T. B., Doesken, N. J., and Kleist, J. 1993. The relationship of drought frequency and duration to time scales, in: Proceedings of the 8th Conference on Applied Climatology, vol. 17, pp. 179–183, American Meteorological Society Boston, MA.

Jackson, C.R., Wang, L., Pachocka, M., Mackay, J.D., and Bloomfield, J.P. 2016. Reconstruction of multi-decadal groundwater level time-series using a lumped conceptual model. Hydrological Processes, 30 (18). 3107-3125. 10.1002/hyp.10850

---

## Author Response (AR1)

Please find below the comments of the editor and the two reviewers. Our replies are in *italics* and preceded by ">>" and the page and line numbers refer to the revised manuscript.

**Editor Jamie Hannaford**

There has been some good discussion on a few key points on this paper, but generally the reviewer comments are minor issues of presentation, clarification and interpretation. The authors suggestions for revisions sound very reasonable and I am confident the final paper will be suitable for publication.

*>> Thanks for this evaluation.*

One issue surrounds the choice of title. I suggested a minor addition at the first editor review stage, but I also agree with John Bloomfield that the title doesn't reflect the paper so well. I can, however, see the merit of the authors' response, i.e. that they wish to keep the link with the companion papers.

In my opinion, it is better for the title to reflect the aims of this paper well rather than force the link with the other two papers. They are quite different in scope, aims and geographical extent. So, personally I would opt for a compromise title that speaks directly to the aims of this paper but does indeed mention the 2015 drought. With this in mind, both new titles suggested by the authors would be suitable and I leave the final choice to them! (for what it's worth I prefer the latter simpler title: "Testing the use of standardised indices....").

*>> Thanks for your thoughts and suggestions for the title of our manuscript. We have changed the title in "Testing the use of standardised indices and GRACE satellite data to estimate the European 2015 groundwater drought in near real-time".*

And of course, highlighting the links between the papers in the text is still a good idea. It is already explained that this paper seeks to provide a groundwater perspective to compare with the other two, but that data availability is on a different level of maturity. I'm sure the reader will appreciate a different approach is required, whereby analysis is first needed to have confidence in these proxy measures, before the 2015 groundwater drought can be fully explored.

I look forward to seeing a revised manuscript.

**Reviewer Miriam Fendekova**

**General comments**

The problem of groundwater level drought evaluation is still less often studied in comparison to meteorological or streamflow droughts. Evaluation of groundwater drought is complicated because of many factors influencing its development and persistence even within relatively small areas. Therefore the paper of authors van Loon et al. is highly appreciated. The paper is well structured,

based on the present-day knowledge of the meteorological and hydrological drought evaluation, and techniques of groundwater drought assessment. The input data, methods and results are well described. The discussion and conclusions are clear, well understandable. The concept of accumulation period estimation and its relation to SPI and SPEI indices led to adequate results within two different areas (southern Germany, Netherlands). Use of satellite-based GRACE-TWS and GRACE-GLDAS showed that because of coarse resolution the models were not able to simulate groundwater anomalies realistically.

*>> Thanks for this positive evaluation of our manuscript.*

**Scientific questions/comments:**

1/ Did the authors think about looking at other groundwater parameters, as baseflow or spring yield from the point of drought occurrence?

*>> Yes, we did consider alternative sources of data such as baseflow or spring yields. For baseflow, data availability would have been higher, because it is derived from discharge time series. Although discharge data availability has its own issues, especially in near-real time, we could have drawn on the FRIEND community effort described in Laaha et al. (2016). However, there would still have been uncertainties as to which method to use to separate baseflow from the total hydrograph and to how representative baseflow time series are for the local groundwater situation during drought. Spring yields are probably one of the best alternatives for groundwater level data to evaluate the groundwater drought situation, but they can only be used in regions where springs occur (i.e. not in the Netherlands and other lowland areas) and the availability and sharing of spring data is even more problematic than that of groundwater level data. We agree that for future research a comparison with baseflow and spring yield data would be very interesting.*

*In the revised manuscript, we have added this suggestion to the Introduction: "At the local scale more sophisticated approaches exist. For example, spring yields are probably one of the best alternatives for groundwater level data to evaluate the groundwater drought situation (Fiorillo, 2009). But these can only be used in regions where springs occur (i.e. not in lowland areas) and the availability and sharing of spring data is even more problematic than that of groundwater level data (Fendeková and Fendek, 2012)." (p.3 l.24-27)*

2/ Hydrogeological conditions and recharge-discharge relationships of an aquifer are more purely represented by spring yields and their changes during the meteorological drought periods. Therefore, maybe the next step in groundwater drought research should be the spring yields drought study. However, in some hydrogeological conditions (as in the Netherlands) the data availability might be very limited.

*>> We totally agree. There is very limited research on spring yields as an indicator for groundwater drought. We are not aware of any databases on spring yield for Europe that could be used in such an analysis, but are very interested to explore this option with Prof. Fendekova in the future.*

3/ Do the authors recommend the use of SPI index which calculation is easier than the SPEI index giving the comparable good results?

*>> Yes, if the data for calculating SPEI are not readily available, one could use SPI without any problem in these or similar regions in Europe. Before application in completely different climates, we do recommend testing first, because in other regions potential evaporation might have more influence on the propagation of drought from meteorological drought to groundwater drought. Also in response to the comments of John Bloomfield about this topic, we have adapted the manuscript to clarify the use of SPI vs. SPEI, see Section 5.1.*

**Technical comments:**

Despite of the reference to Kumar et al. (2016) I would appreciate at least the very short description of major differences in hydrogeological conditions of both areas used in the study. There are not comments to the English language which is excellent, and to figures quality.

*>> In the revised manuscript we expanded the description of the hydrogeological conditions of the two study regions so that the paper is more stand-alone. We added: "Both regions have an oceanic climate (Köppen-Geiger: Cfb), with the southern Germany region experiencing less moderating influence from the ocean. The study area in the Netherlands is composed of sedimentary deposits with varying thickness and composition, ranging from low-lying river valleys with clay and loam sediments and shallow groundwater, to ice-pushed ridges consisting of coarse sand and gravel with an unsaturated zone thickness of up to 30m. The study area in Germany has a large variability in aquifers, due to a hilly to mountainous terrain and a wide range of unconsolidated and consolidated geological formations (Kumar et al., 2016)." (p.5 l.3-8)*

**Reviewer John Bloomfield**

**General comments**

The paper was enjoyable to read, is clearly written and addresses an important set of related research questions on the topic of the predictability of heterogeneous groundwater systems to drought under conditions of incomplete information. My general comments on the paper relate to two points: a.) the framing of the paper, i.e. need for near real-time groundwater drought data and the ability to develop calibrated modelling systems in advance of groundwater droughts, and b.) the title of the paper. As an aside, the authors should be congratulated on seeking to publish a negative finding (their observation that, with regard to the present study, GRACE –TWS did not appear to be suitable for use in real-time groundwater drought assessment) – such findings can be very useful, but are often under-reported.

*>> Thanks for your positive evaluation of our manuscript and your comments on the negative findings.*

**The need for near real-time groundwater drought data.** The paper explores two possible approaches to estimating, in near real-time, spatially varying groundwater levels under conditions of drought by: i.) establishing correlations between a standardised hydrological index (in this case SPEI, although SPI "showed similar results") and observed groundwater levels, and ii.) estimating

groundwater anomalies by subtracting modelled surface water stores from GRACE-TWS data (that represents variability in both groundwater and near surface water). Once calibrated or established such approaches could in theory inform near real-time decisions on groundwater management during future droughts in the absence of groundwater level observations.

In this context, the authors note in their conclusions the following: "We recognise that our approach of using the pre-determined relationship between meteorological conditions and observed groundwater levels is crude and has uncertainties. It does, however, provide a first-order look into the spatio-temporal patterns of current and recent groundwater droughts based on meteorological indices". An alternative approach to modelling the spatial variability of groundwater droughts in near real-time using the available a priori meteorological and groundwater level data could have been developed. Simple lumped parameter models are increasingly used to model groundwater levels; make almost no assumptions about hydrogeological setting; and, if there are sufficient sites, capture and reflect spatially varying responses of groundwater systems. Lumped parameter groundwater models are already used for operational hydrological services, such as UK Hydrological Outlooks http://www.hydoutuk.net/methods/groundwater/. In addition it is possible to constrain the uncertainty in such models as well as the success of their predictions. For example, see work using such models from the UK (Mackay et al., 2014; 2015; Jackson et al., 2016; Marchant et al., 2016). The paper would have a more rounded context if the introduction includes a discussion of such lumped parameter models and the pros and cons of the adopted approach compared with a lumped parameter modelling approach.

*>> We agree that lumped parameter groundwater models could be used in a similar way as our statistical relationship between SPEI and SGI to provide near-real time indications of groundwater drought. In the revised manuscript we have included a discussion of lumped parameter groundwater models in the introduction and its pro and cons in comparison to the approach we used (p.3 l.24-35).*

It should be noted that the SPEI/SGI correlation approach described in the paper produces a calibrated but un-validated correlation and that any SGI values modelled using SPEI driving data will have unconstrained uncertainties. Whereas, using the same calibration and driving data, if multiple lumped parameter groundwater level models are produced they can be both calibrated and validated and uncertainties estimated for each of the predictions of groundwater levels. The main 'cost' in this latter case would be the time involved in producing multiple individual calibrated lumped-parameter models although the process can to some extent be automated.

*>> Our approach has been validated but only qualitatively for the 2003 drought event, by comparing the calculated GW drought condition (Figure 6 and 7) to the observed 'gridded' GW drought condition (Figure 5). We agree that our approach does not allow for a detailed calculation of the uncertainties, which would be possible with lumped parameter groundwater models. Besides the 'cost' of the time investment mentioned by the reviewer, we also want to point out the need for some (but limited) understanding of modelling and groundwater processes needed to calibrate multiple lumped parameter groundwater models. Since our approach is only based on simple statistical correlation, this would increase applicability of the approach in drought monitoring and management. We included these considerations in the introduction of our revised manuscript (p.3 l.24-35).*

The authors also note in their conclusion: "With this work, however, we also want to promote more long-term groundwater measurement and international sharing of groundwater level data". I entirely agree with this statement. For example, throughout Europe groundwater levels and spring discharges are extensively monitored by a wide range of organisations and institutions for a variety of purposes. Some of this information is freely available on the web, however, much of it is not readily available and certainly not in near real-time. Significant advances in the effective management of groundwater resources during droughts could be achieved with better co-ordination and sharing of groundwater data at the European scale. Such a freeing-up of information would in one step obviate the need to model 'near real-time' groundwater levels as described in the current paper and would enable more effective modelling of 'near future' groundwater levels using more sophisticated lumped parameter-type models.

>> *We could not agree more with these statements and we would happily contribute to initiatives that work towards freeing-up of groundwater data.*

**Title of the paper.** The paper aims to establish an approach for estimating near real time groundwater levels during episodes of groundwater drought in the absence of groundwater observations. It uses the expression of the European drought of 2015 from two regions, in Germany and the Netherlands, to test two alternative modelling approaches. However, I don't feel that it is appropriate to suggest that it provides a coherent insight into the groundwater aspects of the European of drought 2015. Consequently, I'd suggest an alternative title such as: "Estimation of near real-time groundwater drought status in the absence of observed groundwater level data".

>> *Thanks for this suggestion. We agree that our paper uses the European 2015 drought as a test case to evaluate the use of two different approaches for near real-time groundwater drought monitoring. However, characterising the 2015 groundwater drought event is an important part of the research that we do want to highlight in the title. Therefore we would like to keep the two-part title, but we changed the order to reflect the primary objective of the work. Based on the suggestion of the reviewer and editor we changed our title to: "Testing the use of standardised indices and GRACE satellite data to estimate the European 2015 groundwater drought in near real-time". Disadvantage of this is that we lose the connection with the two other papers about the 2015 drought event, i.e. Ionita et al. (2016: "The European 2015 drought from a climatological perspective") and Laaha et al. (2016: "The European 2015 drought from a hydrological perspective"), that were also published in the same HESS special issue.*

**Scientific questions/comments:**

P3., last para – given my comments above regarding the title of the paper, I don't think that the statement "In this paper, we aim to analyses the 2015 groundwater drought in Europe . . ." is quite right. I suggest re-phrasing to something like "In this paper, we asses two alternative approaches to model near real-time groundwater drought . . ."

>> *We have slightly rephrased this sentence to focus more on the methodology and less on the 2015 drought event. The sentence now reads: "In this paper, we aim to evaluate two methods that make use of available data to monitor groundwater drought in near real-time on a large scale and test their ability to capture required spatial variability ensuring their local usefulness." (p.4 l.19-20).*

P4., last para - when working with standardised indices such as SPI or SGI it is common practice to produce standardised values on a common time period and with a minimum record length of 30 years (McKee et al 1993 and others). What errors have been introduced into the analysis due to differences in record lengths within and between the two study regions and do these errors effect the conclusions of the study given that "The length of [the groundwater level] records varied from well to well with a minimum of 10 years, starting from the year 1951 for the German wells and 1988 for the Dutch wells and ending in the year 2013"?

*>> We are aware of the advice to use a minimum record length of 30 years and our failure to comply with this. In our previous paper using these groundwater drought observations (Kumar et al. 2016), we quantified the error in the SPI accumulation period and max. correlation caused by the varying record lengths (see figure below). This was what we wrote about it:*

*"We also tested the reliability of the above results against the data availability issue. The A [optimum accumulation period] and rm [max. correlation] obtained across all wells were grouped into three categories according to their available record lengths (i.e., into 10–20, > 20–30, and > 30 years). Both the spread and the average behavior of the optimal accumulation period (A) and the maximum correlation (rm) were comparable across the group of wells with different record lengths (Fig. 3c and d). This shows that the above-presented results are reliable and are not contingent on the selection of wells with either short or long record lengths."*

*Based on the similarities in the data used for this paper, we assume that equally in this work there is no effect of record length on optimum accumulation period between SPEI and SGI. This assumption is now included in Section 2 (p.5 l.26-28).*

[Figure]

*Figure. Box-and-whisker plots of the optimal accumulation period A (top) and the maximum correlation rm (bottom) estimated for a group of wells with varying depth to water tables (left: a), aquifer hydraulic conductivity classes (middle: b), and record lengths (right: c). Results shown for the aquifer hydraulic conductivity corresponding to the German wells are grouped into four distinct classes: high (> 10 -3 ms-1), medium (10 -3–10 -5 ms-1), low (10 -5–10 -7 ms-1), and very low (< 10 -7 ms-1). The percentage of wells falling within each group is indicated at the top of every plot (from Kumar et al., 2016)*

P5-6. Section 3.1 and Fig 2 (top and middle panels) – the SGI data is the same as Kumar et al. (2016) and the difference between Fig 2 of Kumar et al. (2016) and Fig 2 (top and middle panels) of this study is that the latter uses a more refined grid for the analysis. What are the implications, if any, of the reduced number of groundwater level time series observations within the smaller grid cells of the present study on the averaging procedure to obtain a representative SGI for each cell?

*>> The refined grid indeed means that fewer groundwater boreholes are used in the determination of optimal accumulation period of SPEI to match SGI. The advantage of that is that less spatial variability in groundwater is averaged out, which is clearly visible when comparing Fig. 2 from Kumar et al. (2016) with Fig.2 of the current manuscript (see both below). The pattern of accumulation periods of the refined grid (this paper) matches much better that of the individual wells (Kumar et al.,*

[Figure]

*2016) than the coarser grid of Kumar et al. (2016). We mention this in the revised manuscript (p.11 l.9-14).*

**Figure 2.** The **(a)** optimal accumulation $A$ (month) and **(b)** lag periods $L$ (month) required to obtain the **(c)** maximum correlation $r_m$ (–) between the SGI and SPI at point and gridded ($0.5°$) scales for German (top) and Dutch (bottom) data sets.

[Figure]

**Figure 2.** The location of groundwater wells and number of groundwater wells in every $0.25°$ grid cell, the optimal accumulation $A$ (month) and the maximum correlation $r_m$ (-) between the gridded SGI and SPEI for German (top panel) and Dutch (middle panel) data sets. The bottom panels show the correspondence of $A$ and $r_m$ estimated from (SPI vs. SGI) and (SPEI vs. SGI).

P6., last para – "To reduce the noise from individual GLDAS model outputs, we use the ensemble mean of the groundwater anomalies in our analysis". It would be nice to have a bit more information on the scale and nature of the noise in the GLDAS model outputs, perhaps scaled as a function of the GRACE-TWS data? Is there any temporal or spatial structure in the noise relevant to the two study areas and the periods of calibration and modelling?

*>> Since soil depth and number of soil layers in the GLDAS models vary, their total column soil moisture has differences. Moreover, due to the differences in the physical processes in the GLDAS models, there are uncertainties in the simulated soil moisture (for more information see Syed et al., 2008). We estimated groundwater anomaly (GRACE-GW) using the GRACE-TWS and GLDAS soil moisture (GLDAS-SM). To cover the range of uncertainty in GRACE-GW, we used soil moisture from all the four GLDAS models. In the previous version of the manuscript, we only performed a qualitative evaluation of GLDAS model noise (see Appendix A). What is clearly visible is that there are large differences per model. Also in many models there are horizontal bands with high or low anomalies, which do not reflect any physical pattern (for example Fig. A4). As suggested by the reviewer, we plotted the range in both GRACE-TWS products and soil moisture simulations to compare uncertainty for the two selected study regions (like in Long et al., 2013). We discuss this figure and the reasons for the uncertainty in Section 5.2 (p.12 l.3-15).*

P8, para 2 - Once an optimal accumulation period has been established for each cell, why has the maximum (point?) correlation between pairs SPEI/SGI of time series been plotted in Fig 2, wouldn't a representative or (grid) average correlation corresponding to the optimal accumulation period be more instructive than the maximum correlation? It would of course be likely to be lower than the reported correlations.

*>> Here we have to point out a small misunderstanding by the reviewer. The maximum correlation corresponds to the grid-average correlation of the optimal accumulation period. The optimal accumulation period was selected based on the highest correlation between the SPEI for a specific grid cell at a specific accumulation period and the SGI of that same grid cell, and that is the correlation that is reported. So the maximum correlation is not a point correlation. We changed the phrasing in the revised manuscript to avoid confusion by other readers. It now reads: "As expected, the maximum correlation (the correlation obtained for the optimal SPEI accumulation period to match **grid-average** SGI) did not show a specific spatial pattern (Fig. 2 - upper right panels)." (p.9 l.9-10)*

P10., para 1 of Section 5.1 – as above, I suggest re-drafting to the first sentence to "We assessed two alternative approaches to model groundwater drought in the absence of . . . . . ."

*>> We rephrased the first sentence to "We assessed two approaches to monitor groundwater drought in the absence of near real-time groundwater level observations." (p.11 l.5).*

P10., para 2 of Section 5.1 – It is stated that "The analysis using SPI instead of SPEI to represent meteorological conditions gave very similar results. This means that precipitation is the main driver of the optimal accumulation period of meteorological conditions to influence groundwater. This may

be different in more arid regions where PET is a more important component in the water balance. For regions similar to the ones we analysed here, we expect that in absence of PET data SPI can be used instead of SPEI". Although not critical to the paper, this is an interesting observation, but I'm not sure that the interpretation is correct. Bloomfield and Marchant (2013) demonstrated that the optimal accumulation period for SPI/SGI correlation scaled as a function of the autocorrelation range of the groundwater level time series (mmax) (Bloomfield and Marchant, 2013, Fig.10), which in turn was shown to be a function of unsaturated zone thickness and log-hydraulic diffusivity of the aquifer (Bloomfield and Marchant, 2013, Fig. 13), i.e. that it was necessary to invoke aquifer and catchment processes responsible for attenuation of meteorological signals to explain the optimal accumulation period. Assuming that similar relationships hold for SPEI/SGI then I don't think that precipitation is the main control on the optimal accumulation period as stated, rather it is catchment and aquifer characteristics. PET would be expected to have a very limited effect on groundwater levels once a drought has been established when the main cause of groundwater decline would be natural groundwater recession due to groundwater discharge in the absence of precipitation. Note that under drought conditions soil moisture deficits are likely to be very high so limiting the effect of PET.

*>> The reviewer is completely right that we made an important mistake in the formulation here. We did not mean that precipitation is the main driver of the accumulation period for SPI/SGI correlation. The aquifer characteristics are of course what drives the differences in accumulation period between sites. What we meant to say is that the delay in groundwater drought occurrence compared to meteorological drought occurrence is similar if you only take precipitation into account vs. including PET. In the abstract we formulated it like this: "The Standardised Precipitation Index (SPI) showed similar results, which point to a limited influence of potential evaporation in determining the period of influence of meteorological conditions on groundwater levels in our study regions." In the revised manuscript we rephrased the sentences about SPI vs. SPEI in the discussion: "The analysis using SPI instead of SPEI to represent meteorological conditions gave very similar results. This means that adding the effect of potential evaporation does not change the relationship between meteorological forcing and groundwater levels; the delay in groundwater drought occurrence compared to meteorological drought occurrence is similar." (p.11 l.20-22)*

*We cannot, however, say for sure that PET would never have an influence on groundwater drought. The issue during drought is lack of recharge, which does not always equate to absence of precipitation. Also, even under dry soil conditions, evapotranspiration can have an important contribution to storage conditions, as was shown for Europe by Teuling et al. (2013). However, this is a very interesting topic that remains to be investigated more and in different climates.*

P12., end of first para of section 6 – again the statement that "The analysis of both SPI (representing accumulated precipitation anomalies) and SPEI (representing accumulated anomalies in precipitation minus potential evaporation) showed similar results, indicating that precipitation is the main driver of the optimal accumulation period of meteorological conditions to influence groundwater in our study areas". See my comments above. I don't think that this interpretation is correct and that the optimal accumulation period is a function of catchment and aquifer characteristics not precipitation.

*>> We rephrased this sentence: "The analysis of both SPI (representing accumulated precipitation anomalies) and SPEI (representing*

*accumulated anomalies in precipitation minus potential evaporation) showed similar results, indicating that the propagation from meteorological drought to groundwater drought in our study areas is unchanged by including potential evaporation." (p.14 l.8-10)*

**References:**

Ionita, M., Tallaksen, L. M., Kingston, D. G., Stagge, J. H., Laaha, G., Van Lanen, H. A. J., Chelcea, S. M., and Haslinger, K.: The European 2015 drought from a climatological perspective, Hydrol. Earth Syst. Sci. Discuss., doi:10.5194/hess-2016-218, in review, 2016.

Kumar, R., Musuuza, J. L., Van Loon, A. F., Teuling, A. J., Barthel, R., Ten Broek, J., Mai, J., Samaniego, L., and Attinger, S.: Multiscale evaluation of the Standardized Precipitation Index as a groundwater drought indicator, Hydrol. Earth Syst. Sci., 20, 1117-1131, doi:10.5194/hess-20-1117-2016, 2016.

Laaha, G., Gauster, T., Tallaksen, L. M., Vidal, J.-P., Stahl, K., Prudhomme, C., Heudorfer, B., Vlnas, R., Ionita, M., Van Lanen, H. A. J., Adler, M.-J., Caillouet, L., Delus, C., Fendekova, M., Gailliez, S., Hannaford, J., Kingston, D., Van Loon, A. F., Mediero, L., Osuch, M., Romanowicz, R., Sauquet, E., Stagge, J. H., and Wong, W. K.: The European 2015 drought from a hydrological perspective, Hydrol. Earth Syst. Sci. Discuss., doi:10.5194/hess-2016-366, in review, 2016.

Long, D., B. R. Scanlon, L. Longuevergne, A.-Y. Sun, D. N. Fernando, and H. Save (2013), GRACE satellites monitor large depletion in water storage in response to the 2011 drought in Texas, Geophys. Res. Lett., 40, 3395–3401, doi:10.1002/grl.50655.

Syed, T. H., J. S. Famiglietti, M. Rodell, J. Chen, and C. R. Wilson (2008), Analysis of terrestrial water storage changes from GRACE and GLDAS, Water Resour. Res., 44, W02433, doi:10.1029/2006WR005779.

Teuling, A. J., A. F. Van Loon, S. I. Seneviratne, I. Lehner, M. Aubinet, B. Heinesch, C. Bernhofer, T. Grünwald, H. Prasse, and U. Spank (2013), Evapotranspiration amplifies European summer drought, Geophys. Res. Lett., 40, 2071–2075, doi:10.1002/grl.50495.